# Transfer Learning for Control Systems via Neural Simulation Relations

## Abstract

Transfer learning is an umbrella term for machine learning approaches that leverage knowledge gained from solving one problem (the source domain) to improve speed, efficiency, and data requirements in solving a different but related problem (the target domain). The performance of the transferred model in the target domain is typically measured via some notion of loss function in the target domain. This paper focuses on effectively transferring control logic from a source control system to a target control system while providing approximately similar behavioral guarantees in both domains. However, in the absence of a complete characterization of behavioral specifications, this problem cannot be captured in terms of loss functions. To overcome this challenge, we use (approximate) simulation relations to characterize observational equivalence between the behaviors of two systems.

Simulation relations ensure that the outputs of both systems, equipped with their corresponding controllers, remain close to each other over time, and their closeness can be quantified *a priori*. By parameterizing simulation relations with neural networks, we introduce the notion of *neural simulation relations*, which provides a data-driven approach to transfer any synthesized controller, regardless of the specification of interest, along with its proof of correctness. Compared with prior approaches, our method eliminates the need for a closed-loop mathematical model and specific requirements for both the source and target systems. We also introduce validity conditions that, when satisfied, guarantee the closeness of the outputs of two systems equipped with their corresponding controllers, thus eliminating the need for post-facto verification. We demonstrate the effectiveness of our approach through case studies involving a vehicle and a double inverted pendulum.

## 1 Introduction

Humans exhibit remarkable capabilities in transferring expertise (Kendler, 1995) across different tasks where performance in one task is significantly better, having learned a related task. *Transfer learning* (Weiss et al., 2016) is a sub-field of AI that focuses on developing similar capabilities in machine learning problems; aimed towards improving learning speed, efficiency, and data requirements. Unlike conventional machine learning algorithms, which typically focus on individual tasks, transfer learning approaches focus on leveraging knowledge acquired from one or multiple *source* domains to improve learning in a related *target* domain (Weiss et al., 2016). Recently, transfer learning has been successfully applied in designing control logic for dynamical systems (Christiano et al., 2016; Salvato et al., 2021; Nagabandi et al., 2018). However, for safety-critical dynamical systems, the design of the control must provide correctness guarantees on its behavior. In this work, we develop a transfer learning approach for control systems, with formal guarantees on behavior transfer, by learning simulation relations that characterize similarity between the source and target domains. We dub these relations *neural simulation relations*.

This work focuses on controller synthesis for continuous-space control systems described by difference equations. Examples of such systems include autonomous vehicles, implantable medical devices, and power grids. The safety-critical nature of these systems demands formal guarantees—such as safety, liveness, and more expressive logic-based requirements—on the behavior of the resulting control. While deploying the classic control-theoretic approaches may not require mathematical model of the system, and use search (Prajna & Jadbabaie, 2004) and symbolic exploration (Tabuada, 2009) to

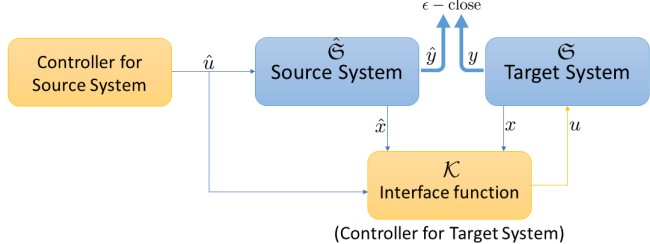

Figure 1: The proposed *behavior transfer* framework for controller synthesis. The existence of a relation and a interface function between source and target systems imply closeness of their behavior.

synthesize controllers, many of these approaches still depend on a mathematical model to provide formal guarantees of correctness. These symbolic approaches typically face the curse-of-dimensionality where the systems with high dimensions become exceedingly cumbersome and time-consuming to design. To overcome these challenges, machine learning based approaches (Zhao et al., 2020; Abate et al., 2022), among others, have been proposed to synthesize control for high-dimensional and complex systems. By making reasonable assumptions (such as Lipschitz continuity) regarding the system, these approaches are able to provide guarantees about their performance. More recently, transfer learning has shown promise (Christiano et al., 2016; Fu et al., 2016; Bousmalis et al., 2018) in transferring controller from a *source domain* (a *low-fidelity* model or a simulation environment) to a *target domain* (*high-fidelity* model or real system). Some of these approaches (Nadali et al., 2023; 2024) also aim to transfer *proof certificates* (such as barrier certificates and closure certificates) in addition to transferring control.

Approaches such as Nadali et al. (2023; 2024) can transfer control and proof certificates when the desired specification on the source and the target system is available. Unfortunately, in typical transfer learning situation, one is interested in transferring a control from a legacy system that is desirable for several, difficult to reify, reasons. In these situations, a formal and complete specification is difficult to extract. We posit that, if we have access to unambiguous structured interfaces (*semantic anchors* (Velasquez, 2023)) between the source and the target environments relating observations in these domains, a behaviorally equivalent transfer can be guaranteed by guaranteeing closeness of these observations as the system evolves in time. We introduce the notion of *Neural Simulation Relations*, which transfers any controller designed for a *source* system, to a *target* system, independent of the property.

In this work we assume access to a simulation environment (digital twin or black-box model) of the source system $\hat{\mathfrak{S}}$. In our proposed behavior transfer approach, as depicted in Figure 1, given a source system $\hat{\mathfrak{S}}$ and a target system $\mathfrak{S}$, we design an interface function $\mathcal{K}$ that can transfer an arbitrary controller from $\hat{\mathfrak{S}}$ to $\mathfrak{S}$. It does so by guaranteeing the existence of an approximate simulation relation $\mathcal{R}$ between the states of the source and target systems such that for any pair of related states, and any input in the source environment, there exists an input in the target environment that keeps the next states related according to $\mathcal{R}$. Moreover, it also guarantees that any pair of states, related via $\mathcal{R}$, have approximately similar observations. The existence of such a simulation relation implies that any behavior on the source system, due to any chosen controller, can be mimicked in the target system. In this work, we train two neural networks to approximate the simulation relation $\mathcal{R}$ and the interface function $\mathcal{K}$. Under reasonable assumptions, we provide validity conditions that, when satisfied, guarantee the closeness of the outputs of two systems equipped with their corresponding controllers, thus eliminating the need for post-facto verification.

**Related Work.** Transfer learning aims at using a previously acquired knowledge in one domain in a different domain. In the context of control systems, transfer learning is typically concerned with transferring a controller from simulation to real-world system which is based on adapting a controller or policy (Fu et al., 2016; Christiano et al., 2016; Bousmalis et al., 2018; Salvato et al., 2021; Nagabandi et al., 2018), or robust control methods that are not affected by the mismatch between the simulator and the real world (Mordatch et al., 2015; Zhou & Doyle, 1998; Berberich et al., 2020). Another approach is to leverage simulation relations (Girard & Pappas, 2011), which is mainly concerned with controlling a complex target system through a simpler source system. The

results in Girard & Pappas (2011; 2009) introduced a sound hierarchical control scheme based on the notion of an *approximate simulation function (relation)*, bringing together theories from both control and automata theory under a unified framework. This relation has had a profound impact on software verification (Baier & Katoen, 2008; Clarke, 1997), synthesizing controllers against logical properties (da Silva et al., 2019; Fainekos et al., 2007; Zhong et al., 2023) across a variety of systems, such as piecewise affine (Song et al., 2022), control affine (Smith et al., 2019; 2020), and descriptor systems (Haesaert & Soudjani, 2020). Additionally, it has been applied in various robotics applications, such as legged (Kurtz et al., 2020) and humanoid (Kurtz et al., 2019) robots. Moreover, Abate et al. (2024) proposed bisimulation learning to find a finite abstract system. More recently, new approaches have been proposed to transfer safety proofs between two similar systems (Nadali et al., 2023; 2024).

**Contribution.** In this paper, we propose sufficient data-driven criteria, dubbed *Neural Simulation Relation*, to ensure transfer of controllers designed for source systems, along with their correctness proofs (if existing), to target systems. In particular, we introduce a training framework that parameterizes the simulation relation function and its associated interface function as neural networks. Furthermore, by proposing validity conditions to ensure the correctness of these functions, we formally guarantee behavioral transfer from a source to a target system, eliminating the need for post-facto verification over the neural networks.

To the best of our knowledge, this is the first correct-by-construction work that aims to find a simulation relation and its interface function in a data-driven manner between two given systems.

## 2 PROBLEM FORMULATION

We denote the set of reals and non-negative reals by $\mathbb{R}$ and $\mathbb{R}_{\geq 0}$, respectively. Given sets $A$ and $B$, $A \setminus B$ and $A \times B$ represent the set difference and Cartesian product between $A$ and $B$, respectively, and $|A|$ represents the cardinality of a set $A$. Moreover, we consider $n$-dimensional Euclidean space $\mathbb{R}^n$ equipped with infinity norm, defined as $\|x - y\| := \max_{1 \leq i \leq n} |x_i - y_i|$ for $x = (x_1, x_2, \ldots, x_n), y = (y_1, y_2, \ldots, y_n) \in \mathbb{R}^n$. Similarly, we denote Euclidean norm as $\|x - y\|_2 := \sqrt{\sum_{i=1}^n (x_i - y_i)^2}$. Furthermore $\langle x_0, x_1, \ldots \rangle, x(k) \in \mathbb{R}^n$ denotes an infinite sequence.

Throughout the paper, we focus on discrete-time control systems (dtCS), as defined below.

**Definition 1.** *A discrete-time control system (dtCS) is a tuple* $\mathfrak{S} := (\mathcal{X}, \mathcal{X}_0, \mathcal{Y}, U, f, h)$, *where* $\mathcal{X} \subseteq \mathbb{R}^n$ *represents the state set,* $\mathcal{X}_0 \subseteq \mathcal{X}$ *is the initial state set,* $U \subseteq \mathbb{R}^m$ *is the set of inputs, and* $\mathcal{Y} \subseteq \mathbb{R}^l$ *is the set of outputs. Furthermore,* $f : \mathcal{X} \times U \to \mathcal{X}$ *is the state transition function, and* $h : \mathcal{X} \to \mathcal{Y}$ *is the output function. The evolution of the system under input from controller* $u : \mathcal{X} \to U$ *is described by:*

$$\mathfrak{S} : \begin{cases} x(t+1) = f(x(t), u(t)), \\ y(t) = h(x(t)), \quad t \in \mathbb{N}, \end{cases} \tag{1}$$

A state sequence of system $\mathfrak{S}$ is denoted by $\langle x_0, x_1, \ldots \rangle$, where $x_0 \in \mathcal{X}_0$, and $x_{i+1} = f(x_i, u_i), u_i \in U$, for all $i \in \mathbb{N}$. We assume that sets $\mathcal{X}$, $U$, and $\mathcal{Y}$ are compact, and maps $f$ and $h$ are unknown but can be simulated via a black-box model. Moreover, we assume that $f$ and $h$ are Lipschitz continuous, as stated in the following assumption.

**Assumption 2** (Lipschitz Continuity). *Consider a dt-CS* $\mathfrak{S} = (\mathcal{X}, \mathcal{X}_0, \mathcal{Y}, U, f, h)$. *The map* $f$ *is Lipschitz continuous in the sense that there exists constants* $\mathcal{L}_u, \mathcal{L}_x \in \mathbb{R}_{\geq 0}$ *such that for all* $x, x' \in \mathcal{X}$, *and* $u, u' \in U$ *one has:*

$$\|f(x, u) - f(x', u')\| \leq \mathcal{L}_x \|x - x'\| + \mathcal{L}_u \|u - u'\|. \tag{2}$$

*Furthermore, the map* $h$ *is Lipschitz continuous in the sense that there exists some constant* $\mathcal{L}_h \in \mathbb{R}_{\geq 0}$ *such that for all* $x, x' \in \mathcal{X}$, *one has:*

$$\|h(x) - h(x')\| \leq \mathcal{L}_h \|x - x'\|. \tag{3}$$

Without loss of generality, we assume that Lipschitz constants of the functions $f$ and $h$ are known. If the Lipschitz constants are unknown, one can leverage sampling methods (Wood & Zhang, 1996; Strongin et al., 2019; Calliess et al., 2020) to estimate those constants. For those dt-CSs satisfying Assumption 2, we define a notion of *behavior transfer*.

**Definition 3.** (Behavior Transfer) *Consider two dt-CSs* $\mathfrak{S} = (\mathcal{X}, \mathcal{X}_0, \mathcal{Y}, U, f, h)$ *(a.k.a.* target system*) and* $\hat{\mathfrak{S}} = (\hat{\mathcal{X}}, \hat{\mathcal{X}}_0, \mathcal{Y}, \hat{U}, \hat{f}, \hat{h})$ *(a.k.a.* source system*), and a constant* $\epsilon \in \mathbb{R}_{\geq 0}$*. There exists a* behavior transfer *from* $\hat{\mathfrak{S}}$ *to* $\mathfrak{S}$ *with respect to* $\epsilon$*, if for any controller for* $\hat{\mathfrak{S}}$*, there exists one for* $\mathfrak{S}$*, such that each systems' output, equipped with their corresponding controllers, remain* $\epsilon$*-close for all time. Concretely, for any state sequence* $\hat{X} = \langle \hat{x}_0, \hat{x}_1, \ldots \rangle$ *in the source system equipped with its controller, there exists a controller and a state sequence* $X = \langle x_0, x_1, \ldots \rangle$ *in the target system equipped with the controller, such that:*

$$\|h(x_t) - \hat{h}(\hat{x}_t)\| \leq \epsilon, \quad \text{for all } t \in \mathbb{N}.$$

Intuitively, if a *behavior transfer* exists from $\hat{\mathfrak{S}}$ to $\mathfrak{S}$, one can adapt any control policy from $\hat{\mathfrak{S}}$ to $\mathfrak{S}$ while ensuring their outputs remain similar (*i.e.,* within $\epsilon$) at all times. To automate the transfer of control strategies in different domains, with theoretical guarantees, this paper aims at solving such a *behavior transfer* from $\hat{\mathfrak{S}}$ to $\mathfrak{S}$, as formally stated in the problem formulation below.

**Problem 4** (Behavior Transfer). *Consider two dt-CSs* $\mathfrak{S} = (\mathcal{X}, \mathcal{X}_0, \mathcal{Y}, U, f, h)$ *(a.k.a.* target system*) and* $\hat{\mathfrak{S}} = (\hat{\mathcal{X}}, \hat{\mathcal{X}}_0, \mathcal{Y}, \hat{U}, \hat{f}, \hat{h})$ *(a.k.a.* source system*), and a constant* $\epsilon \in \mathbb{R}_{\geq 0}$*. Verify a behavior transfer from* $\hat{\mathfrak{S}}$ *to* $\mathfrak{S}$ *with respect to* $\epsilon$ *(if existing).*

To solve Problem 4, the notion of $\epsilon$-*approximate simulation relation* is deployed throughout the paper, which is recalled from Tabuada (2009) as the following.

**Definition 5** (Approximate Simulation Relation). *Consider two dtCSs* $\mathfrak{S} = (\mathcal{X}, \mathcal{X}_0, \mathcal{Y}, U, f, h)$ *and* $\hat{\mathfrak{S}} = (\hat{\mathcal{X}}, \hat{\mathcal{X}}_0, \mathcal{Y}, \hat{U}, \hat{f}, \hat{h})$*, and a constant* $\epsilon \in \mathbb{R}_{\geq 0}$*. A relation* $\mathcal{R} \subseteq \mathcal{X} \times \hat{\mathcal{X}}$ *is an* $\epsilon$*-approximate simulation relation from* $\hat{\mathfrak{S}}$ *to* $\mathfrak{S}$ *if following conditions hold:*

$$(i) \ \forall \hat{x}_0 \in \hat{\mathcal{X}}_0, \exists x_0 \in \mathcal{X}_0 \text{ such that } (x_0, \hat{x}_0) \in \mathcal{R}, \tag{4}$$

$$(ii) \ \forall (x, \hat{x}) \in \mathcal{R}, \text{ we have that } \|h(x) - \hat{h}(\hat{x})\| \leq \epsilon, \tag{5}$$

$$(iii) \ \forall (x, \hat{x}) \in \mathcal{R} \text{ and } \forall \hat{u} \in \hat{U}, \text{ we have that } \exists u \in U \text{ such that } (f(x, u), \hat{f}(\hat{x}, \hat{u})) \in \mathcal{R}. \tag{6}$$

*The system* $\hat{\mathfrak{S}}$ *is said to be* $\epsilon$*-approximately simulated by* $\mathfrak{S}$*, denoted by* $\hat{\mathfrak{S}} \preceq^\epsilon \mathfrak{S}$*, if there exists an* $\epsilon$*-approximate simulation relation from* $\hat{\mathfrak{S}}$ *to* $\mathfrak{S}$*.*

Note that condition (6) tacitly implies the existence of an interface function $\mathcal{K} : \mathcal{X} \times \hat{\mathcal{X}} \times \hat{U} \to U$ as in Figure 1, which acts as transferred controller for $\mathfrak{S}$. The next proposition shows that one can solve Problem 4 by searching for an $\epsilon$-*approximate simulation relation* from $\hat{\mathfrak{S}}$ to $\mathfrak{S}$ (if existing).

**Proposition 6** (Approximate Simulation Relation Imply Transferability(Tabuada, 2009))**.** *Consider two dtCSs* $\mathfrak{S} = (\mathcal{X}, \mathcal{X}_0, \mathcal{Y}, U, f, h)$ *and* $\hat{\mathfrak{S}} = (\hat{\mathcal{X}}, \hat{\mathcal{X}}_0, \mathcal{Y}, \hat{U}, \hat{f}, \hat{h})$*, and a constant* $\epsilon \in \mathbb{R}_{\geq 0}$*. If there exists an* $\epsilon$*-approximate simulation relation from* $\hat{\mathfrak{S}}$ *to* $\mathfrak{S}$ *as in Definition 5, then there exists a behavior transfer from* $\hat{\mathfrak{S}}$ *to* $\mathfrak{S}$ *with respect to* $\epsilon$ *as in Definition 3.*

From this proposition, Problem 4 reduces to the search of an $\epsilon$-approximate simulation relation $\mathcal{R}$ from $\hat{\mathfrak{S}}$ to $\mathfrak{S}$ alongside its associated interface function $\mathcal{K}$. To circumvent the need for mathematical models of $\hat{\mathfrak{S}}$ and $\mathfrak{S}$ and enable the discovery of $\mathcal{R}$ through their black-box representations, we learn the relation $\mathcal{R}$ and the interface function $\mathcal{K}$ as neural networks (Goodfellow et al., 2016).

**Definition 7.** *A neural network with* $k \in \mathbb{N}$ *layers is a function* $F : \mathbb{R}^{n_i} \to \mathbb{R}^{n_o}$*, which computes an output* $y_k \in \mathbb{R}^{n_o}$ *for any input* $y_0 \in \mathbb{R}^{n_i}$ *such that* $y_j = \sigma(W_j y_{j-1} + b_j)$*, with* $j \in \{1, \ldots, k\}$*, where* $W_j$ *and* $b_j$ *are weight matrix and bias vectors, respectively, of appropriate sizes, and* $\sigma$ *is the activation function. Additionally,* $y_{j-1}$ *and* $y_j$ *are referred to as the input and output of the* $j$*-th layer, respectively.*

In this paper, we consider neural networks with ReLU activation function, defined as $\sigma(x) := \max(0, x)$. Such networks describe Lipschitz continuous functions, with Lipschitz constant $\mathcal{L}_F \in \mathbb{R}_{\geq 0}$, in the sense that for all $x_1', x_2' \in \mathbb{R}^{n_i}$, the following condition holds:

$$\|F(x_1') - F(x_2')\| \leq \mathcal{L}_F \|x_1' - x_2'\|. \tag{7}$$

Note that one can obtain an upper bound for Lipschitz constant of a neural network with ReLU activations using spectral norm (Combettes & Pesquet, 2020).

In the next section, considering Proposition 6, we propose a data-driven approach to learn a neural-network-based approximate simulation relation (referred to as *neural simulation relation*) from a source system $\hat{\mathfrak{S}}$ to a target system $\mathfrak{S}$ to solve Problem 4.

## 3 NEURAL SIMULATION RELATIONS

In this section, we focus on how to train neural networks to construct a so-called neural simulation relation (cf. Definition 8) from a source system to a target system to solve Problem 4. To this end, we first introduce the construction of the dataset for training these networks. Based on condition (5), one may observe that for any $(x, \hat{x}) \in \mathcal{R} \subseteq \mathcal{X} \times \hat{\mathcal{X}}$, outputs $h(x)$ and $\hat{h}(\hat{x})$ should be $\epsilon$-close. Therefore, to train neural networks, we only consider those $\mathcal{R} \subseteq \mathcal{T}$, with $\mathcal{T}$ being defined as:

$$\mathcal{T} := \{(x, \hat{x}) \in \mathcal{X} \times \hat{\mathcal{X}}, \mid \|h(x) - \hat{h}(\hat{x})\| \leq \epsilon\}. \tag{8}$$

Then, to construct the data sets with finitely many data points, we cover the set $\mathcal{T}$ by finitely many disjoint hypercubes $\mathcal{T}_1, \mathcal{T}_2, \ldots, \mathcal{T}_M$, by picking a discretization parameter $\mathfrak{e} > 0$, such that:

$$\|t - t_i\| \leq \frac{\mathfrak{e}}{2}, \text{ for all } t \in \mathcal{T}_i, \tag{9}$$

where $t_i$ is the center of hypercube $\mathcal{T}_i$, $i \in \{1, \ldots, M\}$. Accordingly, we pick the centers of these hypercubes as sample points, and denote the set of all sample points by $\mathcal{T}_d := \{t_1, \ldots, t_M\}$. Moreover, consider a hypercube $\hat{\mathcal{X}}_0^{\text{over}}$ over-approximating $\hat{\mathcal{X}}_0$ (i.e., $\hat{\mathcal{X}}_0 \subseteq \hat{\mathcal{X}}_0^{\text{over}}$), and a hypercube $\mathcal{X}_0^{\text{under}}$ under-approximating $\mathcal{X}_0$ (i.e., $\mathcal{X}_0^{under} \subseteq \mathcal{X}_0$). We discretize $\hat{U}, \hat{\mathcal{X}}_0^{\text{over}}$, and $\mathcal{X}_0^{\text{under}}$ in the same manner with discretization parameters $\hat{\mathfrak{e}}, \mathfrak{e}, \mathfrak{e}$, resulting in data sets $\hat{U}_d, \hat{\mathcal{X}}_0^d$, and $\mathcal{X}_0^d$, respectively. Note that we need an *over-approximation* of $\hat{\mathcal{X}}_0$ and an *under-approximation* of $\mathcal{X}_0$ to ensure that *for all $\hat{x}_0 \in \hat{\mathcal{X}}_0^d \subset \hat{\mathcal{X}}_0$, there exists $x_0 \in \mathcal{X}_0^d \subset \mathcal{X}_0$ such that 10 holds.*

Having these data sets, we are ready to introduce the notion of *neural simulation relation* and its associate interface function, which will be trained using these data.

**Definition 8.** *Consider two dtCSs, $\hat{\mathfrak{S}}=(\hat{\mathcal{X}}, \hat{\mathcal{X}}_0, \mathcal{Y}, \hat{U}, \hat{f}, \hat{h})$ (a.k.a source system) and $\mathfrak{S}=(\mathcal{X}, \mathcal{X}_0, \mathcal{Y}, U, f, h)$ (a.k.a target system), a constant $\epsilon \in \mathbb{R}_{\geq 0}$, and neural networks $V : \mathcal{X} \times \hat{\mathcal{X}} \to [0,1]$ and $\mathcal{K} : \mathcal{X} \times \hat{\mathcal{X}} \times \hat{U} \to U$. A relation $\mathcal{R}_{dd}:=\{(x, \hat{x}) \in \mathcal{X} \times \hat{\mathcal{X}} \mid V(x, \hat{x}) \geq 0.5, \|h(x) - \hat{h}(\hat{x})\| \leq \epsilon\}$, is called a neural simulation relation from $\hat{\mathfrak{S}}$ to $\mathfrak{S}$ with the associated interface function $\mathcal{K}$, if the following conditions hold:*

$$\forall \hat{x}_0 \in \hat{\mathcal{X}}_0^d, \exists x_0 \in \mathcal{X}_0^d \text{ such that } V(x_0, \hat{x}_0) \geq 0.5 + \eta, \tag{10}$$

$$\forall (x, \hat{x}) \in \mathcal{T}_d \text{ such that } \|h(f(x, \mathcal{K}(x, \hat{x}, \hat{u}))) - \hat{h}(\hat{f}(\hat{x}, \hat{u}))\| < \epsilon - \gamma \to V(x, \hat{x}) \geq 0.5 + \eta, \tag{11}$$

$$\forall (x, \hat{x}) \in \mathcal{T}_d \text{ such that } \|h(f(x, \mathcal{K}(x, \hat{x}, \hat{u}))) - \hat{h}(\hat{f}(\hat{x}, \hat{u}))\| \geq \epsilon - \gamma \to V(x, \hat{x}) < 0.5 - \eta, \tag{12}$$

$$\forall (x, \hat{x}) \in \mathcal{T}_d \text{ such that } V(x, \hat{x}) \geq 0.5 + \eta \to V(f(x, \mathcal{K}(x, \hat{x}, \hat{u})), \hat{f}(\hat{x}, \hat{u})) \geq V(x, \hat{x}) + \eta, \tag{13}$$

*where $\eta, \gamma \in \mathbb{R}_{>0}$ are some user-defined robustness parameters, and $\hat{u} \in \hat{U}_d$.*

We opted to employ a classifier network for simulation function $V$, since we faced numerical instabilities with the classical definition (Girard & Pappas, 2009).

Each condition of neural simulation relation, is closely related to Definition 5. Condition 10 corresponds to condition 4, conditions 11 and 12 correspond to condition 5, and condition 13 corresponds to condition 6. In order to obtain a neural simulation relation and its associated interface function $\mathcal{K}$ satisfying 10-13, we train the network $V$ with loss $l := l_1 + l_2 + l_3 + l_4$, in which

$l_1 := CE(V(x, \hat{x}), 1), \quad \forall (x, \hat{x}) \in X_{nsi},$

$l_2 := CE(V(x, \hat{x}), 1), \ \forall (x, \hat{x}) \in \mathcal{T}_d, \forall \hat{u} \in \hat{U}_d \text{ such that } \|h(f(x, \mathcal{K}(x, \hat{x}, \hat{u}))) - \hat{h}(\hat{f}(\hat{x}, \hat{u}))\| < \epsilon - \gamma.$

$l_3 := CE(V(f(x, \mathcal{K}(x, \hat{x}, \hat{u})), \hat{f}(\hat{x}, \hat{u})), 1), \ \forall (x, \hat{x}) \in \mathcal{T}_d, \forall \hat{u} \in \hat{U}_d \text{ such that } V(x, \hat{x}) \geq 0.5 + \eta.$

$l_4 := CE(V(x, \hat{x}), 0), \forall (x, \hat{x}) \in \mathcal{T}_d, \forall \hat{u} \in \hat{U}_d \text{ such that } \|h(f(x, \mathcal{K}(x, \hat{x}, \hat{u}))) - \hat{h}(\hat{f}(\hat{x}, \hat{u}))\| \geq \epsilon - \gamma,$

where $CE(x, y) := y \log x + (1 - y) \log(1 - x)$ is the cross-entropy loss function, suited for training classifier networks (Goodfellow et al., 2016). Specifically, $l_1$ encodes the condition (10), and $X_{nsi}$ denotes the set of all points that violate condition (10). Losses $l_2$ and $l_4$ encode conditions (12) and (11) for the network $V$, respectively, $l_3$ encodes condition (13). Additionally, we train the network $\mathcal{K}$ employing the following loss

$$l_k := MSE(h(f(x, \mathcal{K}(x, \hat{x}, \hat{u}))), \hat{h}(\hat{f}(\hat{x}, \hat{u}))), \; \forall (x, \hat{x}) \in \mathcal{T}_d \text{ such that } V(x, \hat{x}) \geq 0.5 + \eta, \; \forall \hat{u} \in \hat{U}_d,$$

where $MSE(x, y) := \frac{1}{n} \sum_{i=1}^{n} \sqrt{(x_i - y_i)^2}$ is the mean squared error loss function (Goodfellow et al., 2016). By leveraging $l_k$, the network $\mathcal{K}$ is trained to produce an input for the target system such that the outputs of target and source systems are $\epsilon$-close at the next time step, regardless of the input provided to the source system.

---

**Algorithm 1** Algorithm for Training a Neural Simulation Relation with Formal Guarantee

---

**Input:** Sets $\mathcal{X}_0, \mathcal{X}, U, \hat{\mathcal{X}}_0, \hat{\mathcal{X}}, \hat{U}$ for target and source systems, respectively, as in Definition 1; discretization parameters $\mathfrak{e}$ for sets $\mathcal{X}, \mathcal{X}_0, \hat{\mathcal{X}}, \hat{\mathcal{X}}_0$ and $\hat{\mathfrak{e}}$ for set $\hat{U}$ as in 9; robustness parameters $\eta \in \mathbb{R}_{>0}$ as in 10 and $\gamma \in \mathbb{R}_{>0}$ as in 11; $\mathcal{L}_x, \mathcal{L}_u, \mathcal{L}_h, \mathcal{L}_{\hat{x}}, \mathcal{L}_{\hat{u}}, \mathcal{L}_{\hat{h}}$ as introduced in Assumption 2; the number of iterations $N$ for training each network; the architecture of the neural networks $V$ and $\mathcal{K}$ as in Definition 7; and the desired $\epsilon \in \mathbb{R}_{\geq 0}$ as in Problem 4.
**Output:** Neural networks $V$ (for encoding the neural simulation relation as in Definition 8) and $\mathcal{K}$ (encoding the interface function $\mathcal{K}$).

    Construct data sets $\mathcal{T}_d, \mathcal{X}_0^d, \hat{\mathcal{X}}_0^d$, and $\hat{U}_d$ according to 9.
    Initialize networks $V$ and $\mathcal{K}$ (Goodfellow et al., 2016).
    $\mathcal{L}_V \leftarrow$ Upper bound of Lipschitz constant of $V$ (Combettes & Pesquet, 2020).
    $\mathcal{L}_K \leftarrow$ Upper bound of Lipschitz constant of $\mathcal{K}$ (Combettes & Pesquet, 2020).
    $Sign \leftarrow$ False
    $i \leftarrow 0$
    **while** Conditions (10)-(16) are not satisfied **do**
        **if** Sign **then**
            Train $V$ with loss $l = l_1 + l_2 + l_3 + l_4$, with $l_1, l_2, l_3,$ and $l_4$ as in 10-13, respectively.
        **else**
            $l_k \leftarrow MSE(h(f(x, \mathcal{K}(x, \hat{x}, \hat{u}))), \hat{h}(\hat{f}(\hat{x}, \hat{u})))$
            Train $\mathcal{K}$ via loss $l_k$ over all data points $\hat{u} \in \hat{U}_d$, and $(x, \hat{x}) \in \mathcal{T}_d$ with $V(x, \hat{x}) \geq 0.5 + \eta$
        **end if**
        $i \leftarrow i + 1$
        **if** exists $n \in \mathbb{N}$ such that $i = nN$ **then**
            $Sign \leftarrow not(Sign)$
        **end if**
        $\mathcal{L}_V \leftarrow$ Upper bound of Lipschitz constant of $V$ (Combettes & Pesquet, 2020).
        $\mathcal{L}_K \leftarrow$ Upper bound of Lipschitz constant of $\mathcal{K}$ (Combettes & Pesquet, 2020).
    **end while**
    Return $V, \mathcal{K}$

---

Note that a neural simulation relation as in Definition 8 is not necessarily a valid $\epsilon$-approximate simulation relation as in Definition 5. Since neural networks are trained on finitely many data points, one requires out of sample guarantees in order to prove correctness.

To address this issue, we propose the following validity conditions, which will be leveraged to show that a neural simulation relation satisfies condition (4)-(6)(cf. Theroem 10).

**Assumption 9.** *Consider two dtCSs, $\hat{\mathfrak{S}} = (\hat{\mathcal{X}}, \hat{\mathcal{X}}_0, \mathcal{Y}, \hat{U}, \hat{f}, \hat{h})$ (a.k.a source system) and $\mathfrak{S} = (\mathcal{X}, \mathcal{X}_0, \mathcal{Y}, U, f, h)$ (a.k.a target system), and two fully connected neural networks $V : \mathcal{X} \times \hat{\mathcal{X}} \to [0, 1]$ and $\mathcal{K} : \mathcal{X} \times \hat{\mathcal{X}} \times \hat{U} \to U$, with ReLU activations, satisfying 10-13. We assume the following validity*

*conditions:*

$$\mathcal{L}_h(\mathcal{L}_x \frac{\mathfrak{e}}{2} + \mathcal{L}_u \mathcal{L}_K \max(\frac{\mathfrak{e}}{2}, \frac{\hat{\mathfrak{e}}}{2})) + \mathcal{L}_{\hat{h}}(\mathcal{L}_{\hat{x}} \frac{\mathfrak{e}}{2} + \mathcal{L}_{\hat{u}} \frac{\hat{\mathfrak{e}}}{2}) \leq \gamma, \tag{14}$$

$$\mathcal{L}_V\big(\max(\mathcal{L}_x, \mathcal{L}_{\hat{x}}) \frac{\mathfrak{e}}{2} + \max(\mathcal{L}_u \mathcal{L}_k \frac{\mathfrak{e}}{2}, \mathcal{L}_{\hat{u}} \frac{\hat{\mathfrak{e}}}{2})\big) \leq 2\eta, \tag{15}$$

$$\mathcal{L}_V \frac{\mathfrak{e}}{2} \leq \eta, \tag{16}$$

*where $\eta, \gamma \in \mathbb{R}_{>0}$ are user-defined parameter as in Definition 8, $\mathcal{T}_d, X_0^d, \hat{U}_d$ are constructed according to 9 with dicretization parameters $\mathfrak{e}, \mathfrak{e}, \hat{\mathfrak{e}}$ respectively, and $\hat{u} \in \hat{U}_d$. Additionally, $\mathcal{L}_V, \mathcal{L}_h, \mathcal{L}_{\hat{h}}, \mathcal{L}_K$ are Lipschitz constants of $V, h, \hat{h},$ and $\mathcal{K}$, respectively (cf 3 and 7), and $\mathcal{L}_x, \mathcal{L}_u$ (resp. $\mathcal{L}_{\hat{x}}, \mathcal{L}_{\hat{u}}$) are Lipschitz constants of the target system (resp. the source system), as defined in 2.*

The intuition behind Assumption 9 lies in leveraging Lipschitz continuity to provide formal guarantees. Since neural networks are trained on a finite set of data points, it is crucial to establish out-of-sample performance guarantees to ensure overall correctness.

Lipschitz continuity enables us to extend guarantees from a finite set of training data to the entire state set. Assumption 9 serves as a condition that facilitates this extension. Specifically, it ensures that if a sample point (used during training) satisfies the simulation relation conditions, then all points within a neighborhood centered at the sample point with radius $\mathfrak{e}$ also satisfy those conditions. This approach forms the theoretical foundation needed to bridge the gap between finite data and overall correctness across the entire state set.

Based on Definition 8 and Assumption 9, in Algorithm 1, we summarize the data-driven construction of a neural simulation relation from the source system to the target system with formal guarantees.

## 4 Formal Guarantee for Neural Simulation Relations

In this section, we propose the main result of our paper. This result shows that a neural simulation relation acquired by using Algorithm 1, conditioned on its termination, is in fact an $\epsilon$-approximate simulation relation, *i.e.* it satisfies conditions (4)-(6) and therefore can be deployed to solve Problem 4.

**Theorem 10.** *Consider two dtCSs, $\hat{\mathfrak{S}} = (\hat{\mathcal{X}}, \hat{\mathcal{X}}_0, \mathcal{Y}, \hat{U}, \hat{f}, \hat{h})$ (a.k.a. the source system), with its Lipschitz constants $\mathcal{L}_{\hat{x}}, \mathcal{L}_{\hat{u}},$ and $\mathcal{L}_{\hat{h}},$ and $\mathfrak{S} = (\mathcal{X}, \mathcal{X}_0, \mathcal{Y}, U, f, h)$ (a.k.a. the target system), with its Lipschitz constants $\mathcal{L}_x, \mathcal{L}_u,$ and $\mathcal{L}_h,$ and a constant $\epsilon \in \mathbb{R}_{>0}$. If there exist neural networks $V$ with a Lipschitz constant $\mathcal{L}_V$ and $\mathcal{K}$ with a Lipschitz constant $\mathcal{L}_K$ that satisfy conditions (10) to (16), with $\mathfrak{e}, \hat{\mathfrak{e}}$ being the discretization parameters for state and input sets, respectively, then one has $\hat{\mathfrak{S}} \preceq^\epsilon \mathfrak{S}$, with the $\epsilon$-approximate simulation relation $\mathcal{R}_{dd} := \{(x, \hat{x}) \in \mathcal{X} \times \hat{\mathcal{X}} | V(x, \hat{x}) \geq 0.5, \|h(x) - \hat{h}(\hat{x})\| \leq \epsilon\}$.*

*Proof.* Since condition (5) is satisfied by construction (cf. 8), we show that conditions (4) and (6) are respected by enforcing conditions (10) to (16). First, we show that condition (4) holds. Consider an arbitrary point $\hat{x}_0 \in \hat{\mathcal{X}}_0$. Based on 9, there exists $\hat{x}_{0_i} \in \hat{\mathcal{X}}_0^d$ such that $\|\hat{x}_0 - \hat{x}_{0_i}\| \leq \frac{\mathfrak{e}}{2}$. According to (10), for any $\hat{x}_{0_i} \in \hat{\mathcal{X}}_0^d$, there exists $x_{0_i} \in \mathcal{X}_0^d \subset \mathcal{X}_0$ such that $V(x_{0_i}, \hat{x}_{0_i}) \geq 0.5 + \eta$. Leveraging the Lipschitz continuity of $V$, one has:

$$V(x_{0_i}, \hat{x}_{0_i}) - V(x_{0_i}, \hat{x}_0) \leq \mathcal{L}_V \frac{\mathfrak{e}}{2} \xLongrightarrow{\text{eq 10}} 0.5 + \eta - V(x_{0_i}, \hat{x}_0) \leq \mathcal{L}_V \frac{\mathfrak{e}}{2} \implies V(x_{0_i}, \hat{x}_0) \geq 0.5 + \eta - \mathcal{L}_v \frac{\mathfrak{e}}{2},$$

where $\mathcal{L}_V$ is the Lipschitz constant of the network $V$. Using (16), one gets $V(x_{0_i}, \hat{x}_0) \geq 0.5$. Thus, for any point $\hat{x}_0 \in \hat{\mathcal{X}}_0$, there exists $x_{0_i}$ such that $(x_{0_i}, \hat{x}_0) \in \mathcal{R}_{dd}$. Therefore condition (4) holds.

Next, we show condition (6) in two steps. Concretely, for any $(x, \hat{x}) \in \mathcal{R}_{dd}$ and $\hat{u} \in \hat{U}$, we show 1) Step 1: $V(f(x, \mathcal{K}(x, \hat{x}, \hat{u})), \hat{f}(\hat{x}, \hat{u})) \geq 0.5$, and 2) Step 2: $\|h(f(x, \mathcal{K}(x, \hat{x}, \hat{u}))) - \hat{h}(\hat{f}(\hat{x}, \hat{u}))\| \leq \epsilon$.

**Step 1**: Consider $X := (x, \hat{x}) \in \mathcal{T}$ to be an arbitrary point. Based on 9, there exists $X_i := (x_i, \hat{x}_i) \in \mathcal{T}_d$ such that $\|X - X_i\| \leq \frac{\mathfrak{e}}{2}$. If $V(X_i) \geq 0.5 + \eta$, by leveraging Lipschitz continuity, one gets:

$$V(X_i) - V(X) \leq \mathcal{L}_V \frac{\mathfrak{e}}{2} \implies 0.5 + \eta - \mathcal{L}_V \frac{\mathfrak{e}}{2} \leq V(X) \xLongrightarrow{\text{eq 16}} V(X) \geq 0.5.$$

Therefore, $X \in \mathcal{R}_{dd}$. Now, if $V(X_i) < 0.5 - \eta$ (note that for any $X_i \in \mathcal{T}_d$, one has either $V(X_i) \geq 0.5 + \eta$ or $V(X_i) < 0.5 - \eta$ according to 11 and 12), again by leveraging Lipschitz continuity, one gets:

$$V(X) - V(X_i) < \mathcal{L}_V \frac{\mathfrak{e}}{2} \implies V(X) < 0.5 - \eta + \mathcal{L}_V \frac{\mathfrak{e}}{2} \xrightarrow{\text{eq 16}} V(X) < 0.5.$$

Thus, $X \notin \mathcal{R}_{dd}$. Therefore, one can conclude that $\mathcal{R}_{dd}$ is equivalent to the set $\{X \in \mathcal{T} | \exists X_i \in \mathcal{T}_d$ with $V(X_i) \geq 0.5 + \eta, \|X - X_i\| \leq \frac{\mathfrak{e}}{2}\}$. Consider any $X \in \mathcal{R}_{dd}$ and $\hat{u} \in \hat{U}$, and the corresponding $X_i \in \mathcal{T}_d$, with $V(X_i) \geq 0.5 + \eta$, $U_i := (\mathcal{K}(X_i, \hat{u}_i), \hat{u}_i)$, with $\hat{u}_i \in \hat{U}_d$, and $U_c := (\mathcal{K}(X, \hat{u}), \hat{u})$, such that $\|X - X_i\| \leq \frac{\mathfrak{e}}{2}$ and $\|\hat{u} - \hat{u}_i\| \leq \frac{\hat{\mathfrak{e}}}{2}$. Let's define $F(X, U_c) := (f(x, \mathcal{K}(x, \hat{x}, \hat{u})), \hat{f}(\hat{x}, \hat{u}))$. Then one gets:

$$V(F(X_i, U_i)) - V(F(X, U_c)) \leq \mathcal{L}_V \|F(X_i, U_i) - F(X, U_c)\|.$$

Based on Assumption 2, one has $\mathcal{L}_V \|F(X_i, U_i) - F(X, U_c)\| \leq \mathcal{L}_V \big(\mathcal{L}_{x'} \|X - X_i\| + \mathcal{L}_{u'} \|U_c - U_i\|\big) \leq \mathcal{L}_V \big(\mathcal{L}_{x'} \frac{\mathfrak{e}}{2} + \mathcal{L}_{u'}\big)$, where $\mathcal{L}_{x'} := \max(\mathcal{L}_x, \mathcal{L}_{\hat{x}})$, $\mathcal{L}_{u'} := \max(\mathcal{L}_u \mathcal{L}_K \frac{\mathfrak{e}}{2}, \mathcal{L}_{\hat{u}} \frac{\hat{\mathfrak{e}}}{2})$. Then, one gets $V(F(X_i, U_i)) - \mathcal{L}_V \big(\mathcal{L}_{x'} \frac{\mathfrak{e}}{2} + \mathcal{L}_{u'}\big) \leq V(F(X, U_c))$. Considering 13, one has:

$$V(F(X, U_c)) \geq V(X_i) + \eta - \mathcal{L}_V \big(\mathcal{L}_{x'} \frac{\mathfrak{e}}{2} + \mathcal{L}_{u'}\big) \geq 0.5 + 2\eta - \mathcal{L}_V \big(\mathcal{L}_{x'} \frac{\mathfrak{e}}{2} + \mathcal{L}_{u'}\big).$$

According to 15, for any $X \in \mathcal{R}_{dd}$ and for any $\hat{u} \in \hat{U}$, there exists $u \in U$ (setting $u = \mathcal{K}(x, \hat{x}, \hat{u})$) such that $V(F(X, U_c)) \geq 0.5$. To show $F(X, U_c) \in \mathcal{R}_{dd}$, we still need to show $\|h(f(x, \mathcal{K}(x, \hat{x}, \hat{u}))) - \hat{h}(\hat{f}(\hat{x}, \hat{u}))\| \leq \epsilon$, which is accomplished by Step 2.

**Step 2**: Consider any $X := (x, \hat{x}) \in \mathcal{R}_{dd}$ and $\hat{u} \in \hat{U}$, and the corresponding $X_i := (x_i, \hat{x}_i) \in \mathcal{T}_d$, with $V(X_i) \geq 0.5 + \eta$, and $\hat{u}_i \in \hat{U}_d$, such that $\|X - X_i\| \leq \frac{\mathfrak{e}}{2}$ and $\|\hat{u} - \hat{u}_i\| \leq \frac{\hat{\mathfrak{e}}}{2}$. One gets:

$$\|h(f(x, \mathcal{K}(x, \hat{x}, \hat{u}))) - \hat{h}(\hat{f}(\hat{x}, \hat{u}))\|$$

$$\leq \|h(f(x, \mathcal{K}(x, \hat{x}, \hat{u}))) - h(x_i, \mathcal{K}(x_i, \hat{x}_i, \hat{u}_i)) + \hat{h}(\hat{f}(\hat{x}_i, \hat{u}_i)) - \hat{h}(\hat{f}(\hat{x}, \hat{u}))\|$$

$$+ \|h(x_i, \mathcal{K}(x_i, \hat{x}_i, \hat{u}_i)) - \hat{h}(\hat{f}(\hat{x}_i, \hat{u}_i))\| \tag{17}$$

$$\leq \|h(f(x, \mathcal{K}(x, \hat{x}, \hat{u}))) - h(x_i, \mathcal{K}(x_i, \hat{x}_i, \hat{u}_i)) + \hat{h}(\hat{f}(\hat{x}_i, \hat{u}_i)) - \hat{h}(\hat{f}(\hat{x}, \hat{u}))\| + \epsilon - \gamma \tag{18}$$

$$\leq \|h(f(x, \mathcal{K}(x, \hat{x}, \hat{u}))) - h(x_i, \mathcal{K}(x_i, \hat{x}_i, \hat{u}_i))\| + \|\hat{h}(\hat{f}(\hat{x}_i, \hat{u}_i)) - \hat{h}(\hat{f}(\hat{x}, \hat{u}))\| + \epsilon - \gamma \tag{19}$$

$$\leq \mathcal{L}_h \big(\mathcal{L}_x \|x - x_i\| + \mathcal{L}_u \mathcal{L}_K \|X_i' - X'\|\big) + \mathcal{L}_{\hat{h}} \big(\mathcal{L}_{\hat{x}} \|\hat{x} - \hat{x}_i\| + \mathcal{L}_{\hat{u}} \|\hat{u} - \hat{u}_i\|\big) + \epsilon - \gamma \tag{20}$$

where $X_i' := (x_i, \hat{x}_i, \hat{u}_i)$, $X' := (x, \hat{x}, \hat{u})$, and $\mathcal{L}_K$ is the Lispchitz constant of network $\mathcal{K}$, in which 17 and 19 are the results of triangle inequality, 18 holds according to 11 and 12, while 20 holds considering Assumption 2. Note that $\|X_i' - X'\| \leq \max(\frac{\mathfrak{e}}{2}, \frac{\hat{\mathfrak{e}}}{2})$, and $\|\hat{u} - \hat{u}_i\| \leq \frac{\hat{\mathfrak{e}}}{2}$, by construction. Continued from 20, one gets:

$$\mathcal{L}_h(\mathcal{L}_x \frac{\mathfrak{e}}{2} + \mathcal{L}_u \mathcal{L}_K \max(\frac{\mathfrak{e}}{2}, \frac{\hat{\mathfrak{e}}}{2})) + \mathcal{L}_{\hat{h}} \big(\mathcal{L}_{\hat{x}} \frac{\mathfrak{e}}{2} + \mathcal{L}_{\hat{u}} \frac{\hat{\mathfrak{e}}}{2}\big) + \epsilon - \gamma \leq \epsilon,$$

which holds according to condition (14).

Combining **Step 1** and **Step 2**, one concludes that condition (6) holds. Thus, $\mathcal{R}_{dd}$ satisfies conditions (4)-(6), therefore, $\mathcal{R}_{dd}$ is an $\epsilon$-approximate simulation relation from $\hat{\mathfrak{S}}$ to $\mathfrak{S}$ with $\mathcal{K}$ being its corresponding interface function. $\square$

**Remark 11.** *Note that in order to implement the interface function $\mathcal{K}$ for the target system $\mathfrak{S}$, one needs to have access to a simulation environment (digital twin or black-box model) of the source system $\hat{\mathfrak{S}}$.*

## 5 EXPERIMENTS

In this section, we demonstrate the efficacy of our proposed method with two case studies. All experiments are conducted on an Nvidia RTX 4090 GPU. In all experiments, networks $V$ and $\mathcal{K}$ are parameterized with 5 hidden layers, each containing 20 neurons for $V$ and 200 neurons for

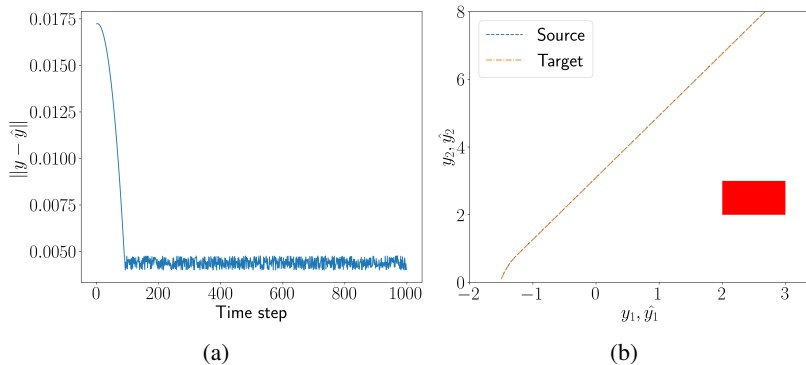

(a)                                    (b)

Figure 2: Figure 2a depicts the error between the outputs, and Figure 2b depicts the trajectories for both systems. Red area depicts the unsafe set.

$\mathcal{K}$, respectively, with ReLU activation for both networks. For both experiments, we train the the networks $V$ and $\mathcal{K}$ according to Algorithm 1 by setting $N = 500$. Although the mathematical models of all systems are reported for simulation purposes, we did not incorporate them to encode neural simulation relations conditions. We have provided a discussion in Appendix 9.1 comparing our method to the state-of-the-art approaches.

### 5.1 VEHICLE

For our first case study, we borrowed vehicle models from Althoff et al. (2017), more details can be found in the Appendix 9.2. The corresponding Lipschitz constants are $\mathcal{L}_x = 1.1, \mathcal{L}_u = 0.1, \mathcal{L}_h = 1, \mathcal{L}_{\hat{x}} = 1.1, \mathcal{L}_{\hat{u}} = 0.1$, and $\mathcal{L}_{\hat{h}} = 1$. We train the networks with the following parameters: $\eta{=}0.3, \gamma{=}0.016, \mathfrak{e}{=}0.002, \hat{\mathfrak{e}}{=}0.0005$, and $\epsilon = 0.02$. In this example, the size of data set is $|\mathcal{T}_d| = 5 \times 10^7$. The training converged in 20 minutes with following parameters: $\mathcal{L}_V{=}23$, and $\mathcal{L}_K{=}8.32 \times 10^{-5}$. The error between outputs of source and target systems over an state sequence of 1000 steps is depicted in Figure 2. Source system is controlled by a safety controller, designed to avoid the unsafe set (Zhao et al., 2020).

### 5.2 DOUBLE PENDULUM

For our second case study, we consider a double inverted pendulum, depicted in Figure 3. Details on target and source systems can be found in the Appendix 9.3. Our algorithm converged in 12 hours with the following parameters: $\mathcal{L}_V{=}13.3, \mathcal{L}_K{=}5.17{\times}10^{-2}, \eta = 0.1, \epsilon = 0.05, \gamma = 0.02, \mathfrak{e} = 0.015, \hat{\mathfrak{e}} = 0.0005$ (resulting in a data set with $|\mathcal{T}_d| = 1.5 \times 10^6$). Output sequences of the source and the target systems are depicted in Figure 3. The source system is controlled by a neural control barrier certificate (Anand & Zamani, 2023), which keeps the pendulum in the upright position.

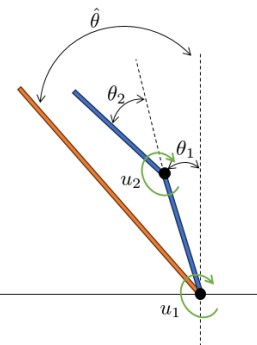

Figure 3: Source and target systems.

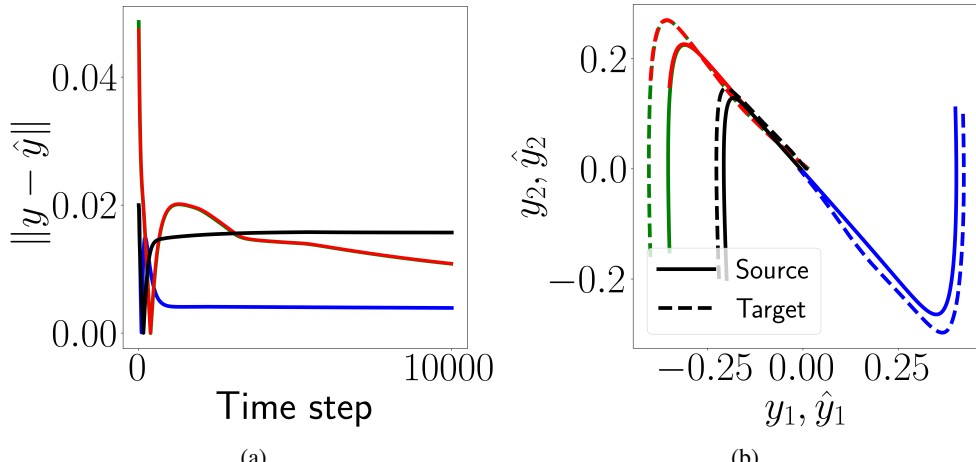

(a)                                                    (b)

Figure 4: Multiple trajectories of target and source systems are depicted in Figure 4b, and their corresponding output error is depicted in Figure 4a.

## 6  CONCLUSION

We proposed a data-driven approach that guarantees the behavior transfer from a source control system to a target control system. We utilized neural networks to find a simulation relation and its corresponding interface function between these systems, we dubbed such relations as *neural simulation relations*. The existence of such functions guarantees that the error between outputs of two systems remain within a certain bound, which enables the behavior transfer. In addition, we propose validity conditions that provide correctness for our neural networks representing the simulation relation and its associated interface function, eliminating the need for post-facto verification. Lastly, we illustrated the effectiveness of our algorithm with two case studies. Possible future direction is to alliviate sample complexity with properties of both target and source systems, such as monotonicity (Angeli & Sontag, 2003) and mixed-monotonicity (Coogan & Arcak, 2015).

## 7  LIMITATIONS

We provided a sufficient condition for transferring controllers between two systems. If our proposed algorithm does not converge, it does not necessarily mean that a simulation relation does not exist. Typically, our algorithm fails to find a simulation relation when the source and target systems are fundamentally different. Additionally, our method is limited by exponential sample complexity, which restricts its applicability to higher-dimensional systems.

## 8  REPEATABILITY STATEMENT

We have outlined details of our proposed method, with its hyper parameters, and the hardware it was trained on in experiments' section. We have also included the code for both case studies in supplementary materials.

## 9  APPENDIX

### 9.1  COMPARISON WITH STATE OF THE ART

To the best of our knowledge, this is the first formally correct result that aims to find a simulation relation and its interface function in a data-driven manner between two given systems. In general, existing works are primarily focused on constructing a source (abstract) system given a target (concrete) system (Abate et al., 2022; 2024; Devonport et al., 2021; Hashimoto et al., 2022). In contrast, our approach does not construct any abstraction. Instead, it establishes a formally correct

transfer of controllers designed for a given abstract (source) system to a concrete (target) system. Methods that aim to find a simulation relation between two given systems typically make restrictive assumptions about the models of both the source and target systems. For example, the results in Zhong et al. (2024) assume linear systems, while Smith et al. (2019) considers only polynomial systems. Furthermore, both methods require access to the mathematical models of the systems. In contrast, our approach makes no assumptions about the specific models of the systems, requiring only access to a black-box representation and the Lipschitz continuity of the dynamics.

## 9.2 EXPERIMENT SETTINGS: VEHICLE

The source and target systems are 3 and 5 dimensional model, respectively. The target system is a 5 dimensional car:

$$x(k+1) = \begin{bmatrix} x_1(k) \\ x_2(k) \\ \delta(k) \\ v(k) \\ \psi(k) \end{bmatrix} + \tau \begin{bmatrix} v(k)\sin(\psi(k)) \\ v(k)\cos(\psi(k)) \\ u_1(k) \\ u_2(k) \\ v(k)\tan(\delta) \end{bmatrix}, y(k) = \begin{bmatrix} 1 & 0 & 0 & 0 & 0 \\ 0 & 1 & 0 & 0 & 0 \end{bmatrix} x(k)$$

where $\tau = 0.1$ is the sampling time, and $x(k) := [x_1(k); x_2(k); \delta(k); v(k); \psi(k)]$ is the state vector, in which $x_1(k), x_2(k), \delta(k), v(k), \psi(k)$ are horizontal position, vertical position, steering angle, velocity, and heading angle at time step $k$, respectively. $u_1(k)$ and $u_2(k)$ are acceleration and steering of the vehicle as control inputs, at time step $k$. The source system is a three dimensional car model, which is used extensively in obstacle avoidance problem (Zhang et al., 2023; Zhao et al., 2020):

$$\hat{x}(k+1) = \begin{bmatrix} \hat{x}_1(k) \\ \hat{x}_2(k) \\ \hat{x}_3(k) \end{bmatrix} + \tau \begin{bmatrix} \sin(\hat{x}_3(k)) \\ \cos(\hat{x}_3(k)) \\ u(k) \end{bmatrix}, \hat{y}(k) = \begin{bmatrix} 1 & 0 & 0 \\ 0 & 1 & 0 \end{bmatrix} \hat{x}(t)$$

where $\hat{x} := [\hat{x}_1, \hat{x}_2, \hat{x}_3]$ is the state vector, in which $\hat{x}_1, \hat{x}_2, \hat{x}_3$ are horizontal position, vertical position, and steering angle, respectively. $u(k)$ is the steering of the vehicle as the control input, at time step $k$.

We consider $\hat{\mathcal{X}} = [-2, 3] \times [0, 8] \times [-1, 1]$, $\hat{\mathcal{X}}_0 = [-2, -1] \times [0, 2] \times [-1, 1]$, $\hat{U} = [-0.5, 0.5]$, which represent the state, initial state and input set of the source system, respectively. Moreover, $\mathcal{X} = \hat{\mathcal{X}} \times [-1, 1]^2$, $\mathcal{X}_0 = \hat{\mathcal{X}}_0 \times [-1, 1]^2$, $U = [-1, 1]^2$, represent the state, initial state and input set of the target system, respectively.

## 9.3 EXPERIMENT SETTINGS: DOUBLE PENDULUM

Target system has the following model:

$$\begin{bmatrix} \theta_1(k+1) \\ \omega_1(k+1) \\ \theta_2(k+1) \\ \omega_2(k+1) \end{bmatrix} = \begin{bmatrix} \theta_1(k) + \tau\omega_1(k) \\ \omega_1(k) + \tau(g\sin(\theta_1(k)) - \sin(\theta_1(k)-\theta_2(k))\omega_1^2(k)) \\ \theta_2(k) + \tau\omega_2(k) \\ \omega_2(k) + \tau(g\sin(\theta_2(k)) + \sin(\theta_1(k)-\theta_2(k))\omega_2^2(k)) \end{bmatrix} + \tau \begin{bmatrix} 0 & 0 \\ 30 & 0 \\ 0 & 0 \\ 0 & 39 \end{bmatrix} \begin{bmatrix} u_1(k) \\ u_2(k) \end{bmatrix};$$

where $[\theta_1(k); \omega_1(k); \theta_2(k); \omega_2(k)] \in [-0.5, 0.5]^4$, and $y(k) = [\theta_1(k), \omega_1(k)]$ is the output. Here, $\theta_1$ and $\theta_2$ represent the angular position of the first and the second joint, respectively, and $\omega_1$ and $\omega_2$ are the angular velocity, respectively, and $u \in [-1, 1]^2$ are the inputs applied to the first and second joint, respectively. The initial set of states are $\mathcal{X}_0 = \mathcal{X}, \hat{\mathcal{X}}_0 = \hat{\mathcal{X}}$ for the target and the source systems, respectively. This is a simplified version of double inverted pendulum, where we assumed the second derivative of both angles are zero, to be able to discretize this system. The source system is an inverted pendulum with the following model:

$$\begin{bmatrix} \hat{\theta}(k+1) \\ \hat{\omega}(k+1) \end{bmatrix} = \begin{bmatrix} \hat{\theta}(k) + \tau\hat{\omega}(k) \\ \hat{\omega}(k) + \tau g\sin(\hat{\theta}(k)) \end{bmatrix} + \tau \begin{bmatrix} 0 \\ 9.1 \end{bmatrix} \hat{u}(k); \ \hat{y}(k) = \begin{bmatrix} 1 & 0 \\ 0 & 1 \end{bmatrix} \hat{x}(k),$$

where $[\hat{\theta}(k); \hat{\omega}(k)] \in [-0.5, 0.5]^2$ represent the angular position and velocity, respectively, and $\tau = 0.01$ is the sampling time, and $\hat{U} = [-1, 1]$ is the input set. Furthermore, for both systems, $g = 9.8$ is the gravitational acceleration. The Lipschitz constants are $\mathcal{L}_x = 1.098, \mathcal{L}_u = 0.39, \mathcal{L}_h = 1, \mathcal{L}_{\hat{x}} = 1.098, \mathcal{L}_{\hat{u}} = 0.091$, and $\mathcal{L}_{\hat{h}} = 1$.

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
