# OpenReview forum: "Transfer Learning for Control Systems via Neural Simulation Relations"
_ICLR.cc/2025/Conference — Submitted to ICLR 2025_

### Official Review · Reviewer_koNa · 2024-10-18

**Soundness:** 3
**Presentation:** 2
**Contribution:** 2
**Rating:** 5
**Confidence:** 4

**Summary:**

The paper focuses on leveraging neural networks to learn simulation relations between two control systems, which enables the transfer of control policies without system models. The proposed approach guarantees that the behaviors of the two systems remain close by learning a relation and an interface function between them. The approach is tested through case studies involving a vehicle model and a double pendulum.

**Strengths:**

1. Using neural networks to learn the similarity between two dynamical systems is interesting.
2. Formal guarantees on simulation are provided.

**Weaknesses:**

1. This work focuses on synthesizing simulation relations, which is a classical problem in formal methods. As this work is positioned as transfer learning, please clarify how it enables transfer learning.
2. While the overall approach is intuitive and theoretically sound, I believe many in-depth analyses are missing.
    - 2.1. As $\epsilon$ should be a small positive number, it usually leads to a very fine-grained partition of the state space in Equation 9 and is supposed to be exponential with the state dimension. It is also reflected by only toy examples considered in the experiment.
    - 2.2. The completeness is not discussed.
3. The experiment only considers toy examples (the vehicle model is 2 dimensional and the double pendulum is 4 dimensional). In addition, the authors do not compare the approach with the state of the arts in simulation/bisimulation learning, e.g., [1] either empirically or theoretically.

Minors:
1. In Definition 3, what does it formally mean by "their corresponding outputs remain $\epsilon$-close for all time"?
2. The figure reference is broken in the first line of Section 5.2.

Overall, this work provides an interesting angle toward controller transferring. However, the current manuscript lacks in-depth discussion, a detailed comparison with STOAs, and a scalability analysis.

[1] Abate, Alessandro, Mirco Giacobbe, and Yannik Schnitzer. "Bisimulation learning." In International Conference on Computer Aided Verification, pp. 161-183. Cham: Springer Nature Switzerland, 2024.

**Questions:**

1. Please clarify how this work enables transfer learning.
2. Please provide a complexity analysis of the discretization approach, especially as it scales with state dimension.
3. Address how the proposed approach might scale to higher-dimensional systems. For instance, the authors can build experiments over any MuJoCo benchmarks to demonstrate the scalability.
4. Please provide a direct comparison with [1] both empirically and theoretically ([1] considers bisimulation relation, the authors can only discuss the simulation part). This work also partitions the state space into finite sets but uses a binary decision tree to learn the simulation relation.

---

> ### Author Response · Authors · 2024-11-20
> **Response to Reviewer koNa (part 1)**
>
> We are grateful for the reviewer’s constructive suggestions and provide our responses below.
>
> Q1.  The completeness is not discussed.
>
> A1. Our method is sound, meaning it provides sufficient conditions for behavior transfer, as highlighted in the contributions. It is important to note that the existence of simulation relations and interface functions for continuous-space systems is not necessary, and to the best of our knowledge, there are no completeness results even in the model-based case.
>
> Q2. The experiment only considers toy examples (the vehicle model is 2 dimensional and the double pendulum is 4 dimensional). In addition, the authors do not compare the approach with the state of the arts in simulation/bisimulation learning, e.g., [1] either empirically or theoretically.
>
> A2. We have included a more realistic case study in the revised version, where the source system is a 3-dimensional car model and the target system is a 5-dimensional car model. The target system:
> \begin{equation}
>    x(k+1) =\left[\begin{array}{c}
>     x_1(k); \\
>     x_2(k); \\
>     \delta (k);\\
>     v(k);
>     \\
>     \psi (k)
>     \end{array}\right]+\tau \left[\begin{array}{c}
>     v(k) \sin (\psi(k)); \\
>     v(k) \cos (\psi(k)); \\
>     u_1(k); \\
>     u_2(k); \\
>     v(k)\text{tan}(\delta)
> \end{array}\right]
> , y(k)=\begin{bmatrix}
>     1 & 0 &0&0&0; \\
>     0 & 1 & 0&0&0
> \end{bmatrix} x(k).\nonumber
> \end{equation}
> The source system:
> \begin{align}
> \hat{x}(k+1)= \left[\begin{array}{c}
>     \hat{x}_1(k); \\
>     \hat{x}_2(k); \\
>     \hat{x}_3 (k)
>     \end{array}\right]+\tau \left[\begin{array}{c}
>      \sin (\hat{x}_3(k)) ;\\
>      \cos (\hat{x}_3(k)) ;\\
>     u(k)
> \end{array}\right]
> , \nonumber
> \hat{y}(k)=\begin{bmatrix}
>     1 & 0 &0 ;\\
>     0 & 1 & 0
> \end{bmatrix}\hat{x}(t).
> \end{align}
> Both models are borrowed from Althof et al., "CommonRoad: Vehicle Models," IEEE Intelligent Vehicles Symposium, 2017.
>
>  To the best of our knowledge, this is the first work that aims to find a simulation relation and its interface function in a data-driven manner between two given systems. Even assuming the availability of the system's model, we attempted prior methods using Sum of Squares (SOS) optimization techniques but were unsuccessful in finding polynomials (up to degree 10) for the double inverted pendulum case study. This computational challenge ultimately led us to abandon increasing the degree of the polynomials further.
>
> In the realm of transfer learning, particularly sim-to-real transfer, existing methods either lack formal guarantees on the correctness of the transfer or are limited to transferring a fixed controller. In contrast, our work provides formal guarantees of $\epsilon$-closeness for the outputs of the source and target systems, ensuring correctness for all controllers designed for the source system when transferred to the target system.
>
> Additionally, we acknowledge the importance of comparing our approach with the state-of-the-art in simulation/bisimulation learning, such as Abate et al., "Bisimulation Learning," CAV 2024, and will include a more detailed discussion and comparison in the revised manuscript.
>
> Q3. Please clarify how this work enables transfer learning.
>
> A3. Our method enables the transfer of previously acquired knowledge (controllers in the source domain) to the target domain, aligning with the principles of transfer learning. This approach is similar to the following sim-to-real transfer learning methods:
>
>
>  Baar et al.,"Sim-to-Real Transfer Learning Using Robustified Controllers in Robotic Tasks Involving Complex Dynamics," ICRA 2019.
>
>  Ushida et al.,"Using Sim-to-Real Transfer Learning to Close Gaps Between Simulation and Real Environments Through Reinforcement Learning," Artificial Life and Robotics, 2021.
>
> Devin et al.,"Learning Modular Neural Network Policies for Multi-Task and Multi-Robot Transfer," ICRA 2017.
>
> Di-Castro et al., "Sim and Real: Better Together," NeurIPS 2021.
>
>
> Our method extends these approaches by providing formal guarantees of $\epsilon$-closeness between the source and target systems, ensuring that controllers designed for the source system remain correct when transferred to the target system.

---

> ### Author Response · Authors · 2024-11-20
> **Response to Reviewer koNa (part 2)**
>
> Q4. Please provide a complexity analysis of the discretization approach, especially as it scales with state dimension.
>
> A4. The sample complexity of our approach is exponential with respect to the state dimension of the system. However, this is a common challenge for data-driven methods that provide formal guarantees and is not unique to our approach.
> Here are some examples to further illustrate this point:
>
> Ansaripour et al., "Learning provably stabilizing neural controllers for discrete-time stochastic systems," ATVA 2023.
>
> Mathiesen et al., "Safety Certification for Stochastic Systems via Neural Barrier Functions," LCSS 2022.
>
>
> Ehlers, "Formal verification of piece-wise linear feed-forward neural network," ATVA 2017.
>
> Elboher et al., "An abstraction-based framework for neural network verification," CAV 2020.
>
>
> Katz et al., "Reluplex: An efficient SMT solver for
> verifying deep neural networks," CAV 2017.
>
> Chang et al., "Neural Lyapunov Control," NeurIPS 2019.
>
>
> Zhao et al.,"Synthesizing barrier certificates using neural networks," HSCC 2020.
>
> Zhang et al., "Exact Verification of ReLU Neural Control Barrier Functions," NeurIPS 2024.
>
>
> That said, we believe our proposed method remains more efficient than many even model-based techniques, as highlighted in our first response to Reviewer y2Rp.
>
> Q5. Address how the proposed approach might scale to higher-dimensional systems. For instance, the authors can build experiments over any MuJoCo benchmarks to demonstrate the scalability.
>
> A5. Our algorithm is not well-suited for high-dimensional systems due to the exponential growth in sample complexity. In our experiments, leveraging 24GB of VRAM on a GPU, we were able to handle a maximum combined dimension of 9 (source and target systems).
>
> We acknowledge the scalability limitations and recognize the importance of exploring ways to extend our approach to higher-dimensional systems. This remains an area of future work, where we aim to incorporate techniques such as dimensionality reduction, compositionality methods, and properties like monotonicity to improve scalability.
>
> Q6. Please provide a direct comparison with [1] both empirically and theoretically ([1] considers bisimulation relation, the authors can only discuss the simulation part). This work also partitions the state space into finite sets but uses a binary decision tree to learn the simulation relation.
>
> A6. We will ensure that the work by Abate et al. [1] is adequately cited in the revised version.
>
> Abate et al. [1] focus on the creation of finite abstractions for concrete systems. In contrast, our approach does not construct any abstraction. While the method in [1] builds an abstraction, our work aims to establish a formally correct transfer of controllers designed for a given abstract (source) system to a concrete (target) system.
>
> Furthermore, the approach in [1] employs a CEGIS loop, which does not scale efficiently to over-parameterized networks. This scalability limitation is a significant difference between their method and ours, as our approach is designed to handle larger networks more effectively. Moreover, CEGIS loop requires access to mathematical model of systems, we assume access to black-box models.
>
> We will expand the discussion in the revised manuscript to provide a detailed theoretical comparison and include relevant empirical insights where applicable.
>
> Q7. Minors
>
> A7.  We appreciate the reviewer's comment. We have included the formal definition of $\epsilon$-closeness in Definition 3, and we have resolved minor typographical errors throughout the paper.
>
> We sincerely thank the reviewer for their detailed analysis and thoughtful feedback on our method. We appreciate the time and effort you have dedicated to reviewing our paper and will incorporate your suggestions to enhance its clarity and quality.
>
> We kindly request that you re-evaluate our paper in light of the clarifications and additional insights we have provided.
>
> Sincerely,
>
> Authors

---

> ### Author Response · Authors · 2024-12-02
>
> With the rebuttal deadline approaching today, we kindly remind the reviewer to review our response to your comments.
>
> If any clarifications or additional details are needed to assist with your review, please feel free to let us know.
>
> Thank you for your time and valuable feedback.

---

### Official Review · Reviewer_59oi · 2024-10-22

**Soundness:** 1
**Presentation:** 1
**Contribution:** 2
**Rating:** 1
**Confidence:** 5

**Summary:**

This paper proposes a data-driven approach that guarantees the behavior transfer from a source control system to a target control system. It also proposes using neural networks to find a simulation relation and its corresponding interface function between these systems, which are called neural simulation relations. The effectiveness of our approach is demonstrated through simple numerical examples of a vehicle and a double-inverted pendulum.

**Strengths:**

The proposed framework can be understood as addressing the domain gap (between the source and target systems) by transfer learning.

**Weaknesses:**

1. As stated in paragraphs 1 and 2 of the Introduction, the safety of a real physical system is a critical motivation behind the proposed work. The design aims to render the target system (e.g., a real system) and source system (e.g., a simulator) have similar behavior, such that the control policy of a source system can be deployed to real systems. A critical limitation is that the controllers of source systems must be verifiable safe; otherwise, you cannot guarantee their safety in target systems. Taking the simulator as one example, a typical challenge is the sim-to-real gap, which hinders the safety guarantee of its controller in the real systems. The proposed approach will also face domain. How can you prove your framework can tolerate such gaps to guarantee safety?
2. Figure 1 describes the framework at a high level. Is it online or offline? However, for each case, I think it will be very hard to apply to real safety-critical systems (such as autonomous vehicles and power grids). If the framework is online, what is the inference time of the interface function? Time is also critical for many safety-critical systems; a second can make a difference. If offline, it can be very hard to apply to the systems (e.g., cars) whose operating environments are dynamic. Safety or system behavior will not be guaranteed in a real-time environment, whose dynamics is not captured by the simulator or offline data-driven training.
3. The reviewer cannot agree with some critical statements in the paper, such as, ``The classic control-theoretic approaches require a mathematical model of the system and use search (Prajna & Jadbabaie, 2004) and symbolic exploration (Tabuada, 2009) to analyze and provide guarantees on the resulting system. These symbolic approaches typically face the curse-of-dimensionality where the systems with high dimensions become exceedingly cumber some and time-consuming to design.” I would suggest the authors have a systematic literature review of classic control (e.g., MPC, linear control, PID) and ML control to figure out why the mathematical model is needed, what unique benefits the mathematical models can offer, and why we need an ML-based or data-driven control.
4. In Definition 3, the formal mathematical formula or definition of ε-close is missing.
5. This paper’s technical contribution is very small. Significant parts of the paper are Definitions, Assumptions, Propositions, etc., cited from others.
6. Definition 8 is very hard to follow and understand. Many more explanations should be included.
7. The motives and explanations for Assumption 9 are missing. Because of this, I cannot conclude the correctness of the theoretical result (Theorem 10) and practice of Assumption 9.
8. The experimental systems are simple and low-dimensional numerical examples. They have a very large mismatch with the real systems. In my opinion, the experimental results are not convincing.
9. The paper lacks comparisons with other transfer learning approaches while considering addressing domain gaps.
10. The paper is not well-written and has many typos (see Figure ?? on page 8).
11. According to Figure 2 (a), I think the paper overstates the proposed framework due to very strong assumptions. I do not believe a control policy for one car can be safely deployed to a very different car (having different weights, different actuator structure, etc.), using the proposed approach.

**Questions:**

1. Why do the conditions in Defination have 0.5? What is the meaning of 0.5?

---

> ### Author Response · Authors · 2024-11-20
> **Response to Reviewer 59oi (part 1)**
>
> We are grateful for the reviewer’s constructive suggestions and provide our responses below.
>
> Q1. As stated in paragraphs 1 and 2 of the Introduction, the safety of a real physical system is a critical motivation behind the proposed work. The design aims to render the target system (e.g., a real system) and source system (e.g., a simulator) have similar behavior, such that the control policy of a source system can be deployed to real systems. A critical limitation is that the controllers of source systems must be verifiable safe; otherwise, you cannot guarantee their safety in target systems. Taking the simulator as one example, a typical challenge is the sim-to-real gap, which hinders the safety guarantee of its controller in the real systems. The proposed approach will also face domain. How can you prove your framework can tolerate such gaps to guarantee safety?
>
> A1. We appreciate the reviewer’s observation regarding the importance of formal correctness guarantees for the source system. However, addressing this aspect lies beyond the scope of our paper.
>
> Our focus, as outlined in Problem 4, is on the transfer of controllers with formal correctness guarantees from the source system to the target system. The design of safety-critical controllers for source systems has been extensively studied, with significant advancements achieved using methods such as barrier certificates. For a comprehensive overview, we refer the reviewer to the works by Ames et al.:
>
> "Control Barrier Functions: Theory and Applications," 18th European Control Conference (ECC).
>
> "Control Barrier Function Based Quadratic Programs for Safety Critical Systems," IEEE Transactions on Automatic Control.
>
> While our method can address gaps between the source and target systems, its primary goal is not limited to this. Synthesizing controllers that satisfy intricate logical properties, such as Linear Temporal Logic (LTL) specifications, is inherently complex. It is often more practical to design such controllers for simplified or low-fidelity models (source systems) and subsequently transfer them to more complex, high-fidelity models (target systems).
>
> Our proposed approach facilitates this transfer while preserving formal guarantees, as demonstrated in our experimental results.
>
> Q2. Figure 1 describes the framework at a high level. Is it online or offline? However, for each case, I think it will be very hard to apply to real safety-critical systems (such as autonomous vehicles and power grids). If the framework is online, what is the inference time of the interface function? Time is also critical for many safety-critical systems; a second can make a difference. If offline, it can be very hard to apply to the systems (e.g., cars) whose operating environments are dynamic. Safety or system behavior will not be guaranteed in a real-time environment, whose dynamics is not captured by the simulator or offline data-driven training.
>
> A2. Our method operates offline. However, training can be conducted in the cloud, and during this process, safety can still be ensured by employing shielding techniques, such as those proposed by Alshiekh et al., in "Safe Reinforcement Learning via Shielding" (AAAI 2018). These techniques allow the system to remain safe while the interface is being designed and deployed onto the system.
>
> Q3. The reviewer cannot agree with some critical statements in the paper, such as, ``The classic control-theoretic approaches require a mathematical model of the system and use search (Prajna & Jadbabaie, 2004) and symbolic exploration (Tabuada, 2009) to analyze and provide guarantees on the resulting system. These symbolic approaches typically face the curse-of-dimensionality where the systems with high dimensions become exceedingly cumber some and time-consuming to design.” I would suggest the authors have a systematic literature review of classic control (e.g., MPC, linear control, PID) and ML control to figure out why the mathematical model is needed, what unique benefits the mathematical models can offer, and why we need an ML-based or data-driven control.
>
> A3.We appreciate the reviewer’s insightful feedback and fully agree with their observation. We will revise the paragraph to ensure greater accuracy. While deploying classical control methods such as PID and MPC may not require an exact mathematical model of the system, many of these approaches still depend on a mathematical model to provide formal guarantees of correctness for the synthesized controllers. It is important to note that formal guarantees of correctness for the transferred controllers are a primary criterion in our work.
>
> Q4. In Definition 3, the formal mathematical formula or definition of $\epsilon$-close is missing.
>
> A4. The reviewer is correct, and we will ensure that the formal definition of
> $\epsilon$-close is included in the revised version.

---

> ### Author Response · Authors · 2024-11-20
> **Response to Reviewer 59oi (part 2)**
>
> Q5. This paper’s technical contribution is very small. Significant parts of the paper are Definitions, Assumptions, Propositions, etc., cited from others.
>
> A5. We respectfully disagree with the reviewer’s assessment. To the best of our knowledge, this is the first work that aims to establish a simulation relation and its corresponding interface function in a data-driven manner between two given systems. It is also the first to use neural networks to represent these components.
>
> Moreover, our proposed notion of simulation relation and interface function introduces, for the first time, a sufficient condition and sound results for the positive transfer of controllers, regardless of the specific properties they are enforcing. Additionally, we provide well-defined conditions that, when satisfied, guarantee the correctness of our Neural Simulation Relation (as demonstrated in Theorem 10).
>
> Q6. Definition 8 is very hard to follow and understand. Many more explanations should be included.
>
> A6. We will ensure that this section is thoroughly revised and clarified in the updated manuscript.
>
> Q7. The motives and explanations for Assumption 9 are missing. Because of this, I cannot conclude the correctness of the theoretical result (Theorem 10) and practice of Assumption 9.
>
> A7. Thank you for highlighting this issue; we will revise the draft to address it. The intuition behind Assumption 9 lies in leveraging Lipschitz continuity to provide formal guarantees. Since neural networks are trained on a finite set of data points, it is crucial to establish out-of-sample performance guarantees to ensure overall correctness.
>
> Lipschitz continuity enables us to extend guarantees from a finite set of training data to the entire state set. Assumption 9 serves as a condition that facilitates this extension. Specifically, it ensures that if a sample point (used during training) satisfies the simulation relation conditions, then all points within a neighborhood centered at the sample point with radius $\mathfrak{e}$ also satisfy those conditions. This approach forms the theoretical foundation needed to bridge the gap between finite data and overall correctness across the entire state set.
>
> Q8. The experimental systems are simple and low-dimensional numerical examples. They have a very large mismatch with the real systems. In my opinion, the experimental results are not convincing.
>
> A8. To the best of our knowledge, this is the first work to establish a simulation relation and its corresponding interface function in a data-driven manner between two given systems. The purpose of the experiments is to demonstrate the theoretical results, but we appreciate your suggestion and will revise the final version to include a more realistic case study. Specifically, we will incorporate a scenario where the source system is a 3-dimensional car and the target system is a 5-dimensional car.
> The target system:
>  \begin{equation}
>    x(k+1) =\left[\begin{array}{c}
>     x_1(k); \\
>     x_2(k); \\
>     \delta (k);\\
>     v(k);
>     \\
>     \psi (k)
>     \end{array}\right]+\tau \left[\begin{array}{c}
>     v(k) \sin (\psi(k)); \\
>     v(k) \cos (\psi(k)); \\
>     u_1(k); \\
>     u_2(k); \\
>     v(k)\text{tan}(\delta)
> \end{array}\right]
> , y(k)=\begin{bmatrix}
>     1 & 0 &0&0&0; \\
>     0 & 1 & 0&0&0
> \end{bmatrix} x(k).\nonumber
> \end{equation}
> The source system:
> \begin{align}
> \hat{x}(k+1)= \left[\begin{array}{c}
>     \hat{x}_1(k); \\
>     \hat{x}_2(k); \\
>     \hat{x}_3 (k)
>     \end{array}\right]+\tau \left[\begin{array}{c}
>      \sin (\hat{x}_3(k)) ;\\
>      \cos (\hat{x}_3(k)) ;\\
>     u(k)
> \end{array}\right]
> , \nonumber
> \hat{y}(k)=\begin{bmatrix}
>     1 & 0 &0 ;\\
>     0 & 1 & 0
> \end{bmatrix}\hat{x}(t).
> \end{align}
> Both models are borrowed from Althof et al., "CommonRoad: Vehicle Models," IEEE Intelligent Vehicles Symposium, 2017.
>
> Moreover, methods that provide formal guarantees typically do not scale well to higher dimensional systems, and this is not unique to our approach. That said, we believe our proposed method is still more efficient than many model-based techniques (as highlighted in our first response to Reviewer y2Rp). Here are some examples to further illustrate this point:
>
> Ansaripour et al., "Learning provably stabilizing neural controllers for discrete-time stochastic systems," ATVA 2023.
>
> Abate et al.,"Stochastic omega-regular verification and control with supermartingales," CAV 2024.
>
> Mathiesen et al., "Safety Certification for Stochastic Systems via Neural Barrier Functions," LCSS 2022.
>
> Elboher et al., "An abstraction-based framework for neural network verification," CAV 2020.
>
> Katz et al., "Reluplex: An efficient SMT solver for verifying deep neural networks," CAV 2017.
>
> Chang et al., "Neural Lyapunov Control," NeurIPS 2019.
>
> Zhao et al.,"Synthesizing barrier certificates using neural networks," HSCC 2020.
>
> Zhang et al., "Exact Verification of ReLU Neural Control Barrier Functions," NeurIPS 2024.

---

> ### Author Response · Authors · 2024-11-20
> **Response to Reviewer 59oi (part 3)**
>
> Q9. The paper lacks comparisons with other transfer learning approaches while considering addressing domain gaps.
>
> A9. To the best of our knowledge, this is the first work that aims to find a simulation relation and its interface function in a data-driven manner between two given systems. Even assuming the availability of the system's model, we attempted prior methods using Sum of Squares (SOS) optimization techniques but were unsuccessful in finding polynomials (up to degree 10) for the double inverted pendulum case study, ultimately leading us to abandon increasing the degree of the polynomials further.
>
> In transfer learning context, there are numerous works on sim-to-real transfer such as
>
> Peng et al., "Sim-to-Real Transfer of Robotic Control with Dynamics Randomization," ICRA 2018.
>
> Anderson et al., "Sim-to-Real Transfer for Vision-and-Language Navigation," CoRL 2021.
>
> Di-Castro et al., "Sim and Real: Better Together," NeurIPS 2021.
>
>
> Generally, methods in transfer learning, either do not provide formal guarantee of correctness, or they are just for transferring a fixed controller. We on the other hand, provide formal guarantee of $\epsilon$-closeness of the outputs, for all controllers designed for the source system.
>
> Q10. The paper is not well-written and has many typos (see Figure ?? on page 8).
>
> A10.  We will ensure that all typos, including the one mentioned (e.g., "Figure ??" on page 8), are fixed in the revised version.
>
> Q11. According to Figure 2 (a), I think the paper overstates the proposed framework due to very strong assumptions. I do not believe a control policy for one car can be safely deployed to a very different car (having different weights, different actuator structure, etc.), using the proposed approach.
>
> A11. It is important to clarify that our method is sound, meaning it provides a sufficient condition for success as long as the training converges. However, the inability to find a simulation relation and interface function using our approach does not imply that such a relation and interface map does not exist. Additionally, we would like to emphasize that the assumptions required by our method are not overly restrictive. This is demonstrated by our ability to successfully showcase results in several case studies, which were previously intractable using existing model-based techniques (cf. our response to your 9th question).
>
> The reviewer is correct that when the gap between the source and target systems is too significant, transferring controllers may not be feasible. In such cases, we suspect our algorithm will fail to converge. To address this, we will include a section discussing failure cases. For example, in our experiments with a DC motor as the source system and an inverted pendulum as the target system, we were unable to identify a simulation relation with formal guarantees, highlighting the limitations of our approach in scenarios involving drastically different systems.
>
> Q12. Why do the conditions in Defination have 0.5? What is the meaning of 0.5?
>
> A12. Since the neural network V serves as a classifier, its output represents the probability of a point belonging to the simulation relation. If this probability is less than 0.5, we assume that the data point does not belong to the simulation relation. While other threshold values are valid, we believe 0.5 is a reasonable and standard choice. For more details on classifier networks and the rationale behind the 0.5 threshold, we refer the reviewer to Goodfellow et al.,"Deep Learning," Chapter 6.
>
> We sincerely thank the reviewer for their detailed analysis and thoughtful feedback on our method. We appreciate the time and effort you have dedicated to reviewing our paper and will incorporate your suggestions to enhance its clarity and quality.
>
> We kindly request that you re-evaluate our paper in light of the clarifications and additional insights we have provided.
>
> Sincerely,
>
> Authors

---

> > ### Comment · Reviewer_59oi · 2024-11-21
> > **Thanks for your response**
> >
> > I appreciate the author's answers to my comments. I will maintain my rating according to the paper's contributions and comments.

---

### Official Review · Reviewer_AenQ · 2024-11-01

**Soundness:** 3
**Presentation:** 2
**Contribution:** 2
**Rating:** 5
**Confidence:** 3

**Summary:**

The paper proposes an approach to establish a type of equivalence between two discrete-time, continuous-state dynamical control systems. Formally, an approximate simulation relation is established between the two systems, which relates states from which trajectories remain within a given distance from each other in the output space, assuming that an appropriate sequence of inputs is applied.

The construction of the simulation relation is achieved indirectly, via two functions: a function V that maps a pair of states of the systems to the interval [0,1], and a function K that maps a pair of states of the systems and an input for the first system to a corresponding input of the second system. While V is used to encode related states, K is used to encode related inputs.

The intended application is to relate states in a complex dynamical system to a surrogate model. If successful, a control policy designed for the simpler surrogate model can then be mapped, using K, to a policy for the complex system.

The basic idea is promising, but there are a number of major issues that remain to be addressed (see weaknesses). In sum, the paper in its current form is not ready for publication.

(review revised after clarification from authors, including an error on my part)


Minor remarks:
- the definition of a control policy is missing (required for Def. 3)
- Prob. 4 is not clearly formulated (*whether* … exists)
- loss on page 5 not clear (how is the forall quantifier implemented)
- line 427: Figure ??

**Strengths:**

The paper proposes to use neural networks to tackle a difficult problem using methods from machine learning.

**Weaknesses:**

- V seems to take the role of a simulation function. It is formulated such that it is amenable to learning using a loss function, but it is not immediately clear why this should work better than a direct approximation of a simulation function.
- The use of the cross entropy loss should be better motivated; it's not immediately clear to me why that would be the best choice.
- The experiments are not convincing. The first one is extremely simple, yet seems to require a surprising amount of computational resources.
- The second experiment constitutes a simplified model of a double pendulum, and it is not immediately clear to me how difficult the problem is. A comparison with a baseline from the literature may help to clarify this issue.
Some of the formal details of the paper are unclear (see minor remarks), and the experiments should be accompanied by more detailed descriptions (training curves, etc.).

**Questions:**

Why did you use cross-entropy loss? Why map V to [0,1] rather than [0,\infty), like a simulation function?

---

> ### Author Response · Authors · 2024-11-20
> **Response to Reviewer AenQ (part 1)**
>
> We are grateful for the reviewer’s constructive suggestions and provide our responses below.
>
> Q1. The experiments are not convincing. The first one is extremely simple, yet seems to require a surprising amount of computational resources.
>
> A1.Thank you for your feedback regarding the first case study. To the best of our knowledge, this is the first work to establish a simulation relation and its corresponding interface function in a data-driven manner between two given systems. The purpose of the experiments is to demonstrate the theoretical results, but we appreciate your suggestion and will revise the final version to include a more realistic case study. Specifically, we will incorporate a scenario where the source system is a 3-dimensional car and the target system is a 5-dimensional car.
> The target system:
> \begin{equation}
>    x(k+1) =\left[\begin{array}{c}
>     x_1(k); \\
>     x_2(k); \\
>     \delta (k);\\
>     v(k);
>     \\
>     \psi (k)
>     \end{array}\right]+\tau \left[\begin{array}{c}
>     v(k) \sin (\psi(k)); \\
>     v(k) \cos (\psi(k)); \\
>     u_1(k); \\
>     u_2(k); \\
>     v(k)\text{tan}(\delta)
> \end{array}\right]
> , y(k)=\begin{bmatrix}
>     1 & 0 &0&0&0; \\
>     0 & 1 & 0&0&0
> \end{bmatrix} x(k).\nonumber
> \end{equation}
> The source system:
> \begin{align}
> \hat{x}(k+1)= \left[\begin{array}{c}
>     \hat{x}_1(k); \\
>     \hat{x}_2(k); \\
>     \hat{x}_3 (k)
>     \end{array}\right]+\tau \left[\begin{array}{c}
>      \sin (\hat{x}_3(k)) ;\\
>      \cos (\hat{x}_3(k)) ;\\
>     u(k)
> \end{array}\right]
> , \nonumber
> \hat{y}(k)=\begin{bmatrix}
>     1 & 0 &0 ;\\
>     0 & 1 & 0
> \end{bmatrix}\hat{x}(t).
> \end{align}
> Both models are borrowed from Althof et al., "CommonRoad: Vehicle Models," IEEE Intelligent Vehicles Symposium, 2017.
>
> We acknowledge the reviewer’s observation about computational expense. While the first experiment converged in 4 minutes, computational complexity is a common challenge for methods that provide formal guarantees, and this is not unique to our approach. That said, we believe our proposed method is still more efficient than many model-based techniques (as highlighted in our first response to Reviewer 1). Here are some examples to further illustrate this point:
>
> Ansaripour et al., "Learning provably stabilizing neural controllers for discrete-time stochastic systems," ATVA 2023.
>
>
> Mathiesen et al., "Safety Certification for Stochastic Systems via Neural Barrier Functions," LCSS 2022.
>
>
> Ehlers, "Formal verification of piece-wise linear feed-forward neural network," ATVA 2017.
>
> Elboher et al., "An abstraction-based framework for neural network verification," CAV 2020.
>
>
> Katz et al., "Reluplex: An efficient SMT solver for
> verifying deep neural networks," CAV 2017.
>
> Chang et al., "Neural Lyapunov Control," NeurIPS 2019.
>
>
> Zhao et al., "Synthesizing barrier certificates using neural networks," HSCC 2020.
>
> Zhang et al., "Exact Verification of ReLU Neural Control Barrier Functions," NeurIPS 2024.
>
>
> Q2.The second experiment is misleading: The equations do not represent the double pendulum shown in Fig. 3, but two uncoupled simple pendulums. Some of the formal details of the paper are unclear (see minor remarks), and the experiments should be accompanied by more detailed descriptions (training curves, etc.).
>
> A2. The model of the double inverted pendulum was derived using the Euler-Lagrange equations, as outlined in the Appendix A.1 of "Course Notes for MIT 6.832" by Tedrake. To enable discretization, we assumed the second derivatives of both angles to be zero. However, it is important to note that the dynamics of the two pendulums are not entirely decoupled, as the term $\sin(\theta_1 - \theta_2)$ introduces coupling in the system's formulation.
>
> Even with this simplified version and assuming access to the exact mathematical model of the system, prior model-based techniques were unable to find a simulation relation and its corresponding interface function, further emphasizing the challenges addressed by our method. We will elaborate on this in the revised version.

---

> > ### Comment · Reviewer_AenQ · 2024-11-26
> >
> > I thank the authors for clarifying many of the points I raised. In particular, I must apologize for missing the coupling terms in the double pendulum model. That said, I maintain that it's important to clarify that this is a simplified model and to indicate its source. A complete double pendulum model is significantly more complex, see for example:
> > Stachowiak T, Okada T. A numerical analysis of chaos in the double pendulum. Chaos, Solitons & Fractals. 2006 Jul 1;29(2):417-22.
> >
> > Concerning related work on constructing a simulation relation from data, the following work may be of use to the authors :
> > - Devonport, A., Saoud, A., & Arcak, M. (2021, December). Symbolic abstractions from data: A PAC learning approach. In 2021 60th IEEE Conference on Decision and Control (CDC) (pp. 599-604). IEEE.
> > - Hashimoto, Kazumune, et al. "Learning-based symbolic abstractions for nonlinear control systems." Automatica 146 (2022): 110646.
> >
> > In response to the author's clarifications, I will raise my score.

---

> ### Author Response · Authors · 2024-11-20
> **Response to Reviewer AenQ (part 2)**
>
> Q3. Why did you use cross-entropy loss? Why map V to [0,1] rather than [0,\infty), like a simulation function? (also first and second weakness mentioned by the reviewer)
>
>
> A3.
> Thank you for raising this excellent point. We will revise the draft to address this potential confusion. In short, we encountered numerical issues with the classical definition of the function $V$ during neural network training, particularly when $\epsilon$ was small. To address this, we slightly modified the definition. It is important to note that the revised conditions still guarantee the closeness of the outputs.
>
> We utilized cross entropy loss, as it is suited for training classifier networks. We refer the reviewer to Goodfellow et al.,``Deep Learning" for more details regarding classifier networks and corss entropy loss.
>
> Additionally, based on our experiments, the most challenging aspect of the process is synthesizing the interface function $\mathcal{K}: \mathcal{X} \times \hat{\mathcal{X}} \times U \to \hat{U}$. By comparison, finding $V$ is relatively straightforward when a suitable $\mathcal{K}$ is provided.
>
> Q4. Minor remarks
>
>
> A4. Since we train neural networks on a finite set of data points, the implementation of quantifiers is straightforward. We then extend the guarantees from the training data to unseen data using Lipschitz continuity, as outlined in conditions (14)-(16). We have also resolved minor typographical errors throughout the paper.
>
> We sincerely thank the reviewer for their detailed analysis and thoughtful feedback on our method. We appreciate the time and effort you have dedicated to reviewing our paper and will incorporate your suggestions to enhance its clarity and quality.
>
> We kindly request that you re-evaluate our paper in light of the clarifications and additional insights we have provided.
>
> Sincerely,
>
> Authors

---

> ### Author Response · Authors · 2024-11-26
>
> We thank the reviewer for raising their score and their suggestions. We make sure there's adequate discussion about suggested work and double inverted pendulum in our final version.
>
> Both papers suggested by the reviewer are concerned with finding an abstraction of a concrete system. In contrast, our approach does not construct any abstraction, our work aims to establish a formally correct transfer of controllers designed for a given abstract (source) system to a concrete (target) system.
>
> Sincerely,
>
> Authors

---

### Official Review · Reviewer_y2Rp · 2024-11-03

**Soundness:** 2
**Presentation:** 2
**Contribution:** 3
**Rating:** 5
**Confidence:** 3

**Summary:**

This paper introduces a way of performing transfer learning by quantifying the closeness of the simulated data among two systems and controllers, along with formal guarantees. The method is evaluated in simple vehicle dynamics and double inverted pendulum.

**Strengths:**

The paper presents the proposed algorithm with rigorous definitions and propositions. The proposed algorithm nicely transforms to reasonable loss functions to train the neural networks $V$ and $\mathcal{K}$. In addition, the author rigorously proved that the resultant neural simulation relation is an $\epsilon$-approximate simulation.

**Weaknesses:**

1. The experiments, though demonstrated $||y - \hat{y}$<\epsilon$ for the observed trajectories and plotted the trajectories, lacks comparison to relevant baselines. Given that the proposed transfer learning algorithm relies on the discretization of the state space, several transfer learning algorithms mentioned in the original paper's reference, even requiring access to the dynamical model, could be compared with by approximating or learning the system dynamics in local regions.
2. The robustness of the trained models with regard to hyper parameters is unclear. An addition of ablation studies on the sensitivity for the hyper parameters could be added.

Nit picks: (1) The figure is not properly referenced at line 427. (2) Grammar error in the sentence of the lines 164-165: "To automate the transfer of control strategies [in] different domains". (3) Line 418-409, "We train the neural networks [with] the following parameters. I suggest the authors to thoroughly read the paper to reduce avoidable errors and increase the clarity of the menuscript.

**Questions:**

I wonder whether the authors have observed failure cases during their experiments. If so, is there a way to measure whether the failure case is due to the difficulty of learning, or due to inherent problems of the given source and target system?

---

> ### Author Response · Authors · 2024-11-20
> **Response to Reviewer y2Rp (part 1)**
>
> We are grateful for the reviewer’s constructive suggestions and provide our responses below.
>
> Q1.The experiments, though demonstrated $|| y-\hat{y}||<\epsilon$ for the observed trajectories and plotted the trajectories, lacks comparison to relevant baselines. Given that the proposed transfer learning algorithm relies on the discretization of the state space, several transfer learning algorithms mentioned in the original paper's reference, even requiring access to the dynamical model, could be compared with by approximating or learning the system dynamics in local regions.
>
> A1.  To the best of our knowledge, this is the first work that aims to find a simulation relation and its interface function in a data-driven manner between two given systems. Even assuming the availability of the system's model, we attempted prior methods using Sum of Squares (SOS) optimization techniques but were unsuccessful in finding polynomials (up to degree 10) for the double inverted pendulum case study, ultimately leading us to abandon increasing the degree of the polynomials further.
>
> In the realm of transfer learning (particularly sim-to-real transfer), existing methods either lack formal guarantees on the correctness of the transfer or are limited to transferring a fixed controller. In contrast, our work provides formal guarantees of $\epsilon$-closeness for the outputs of the source and target systems, ensuring correctness for all controllers designed for the source system when transferred to the target system.
>
> We will revise the draft to include a more realistic scenario inspired by a car example (borrowed from literature), where the source system is a 3-dimensional car model, and the target system is a 5-dimensional car model. The target system:
> \begin{equation}
>     x(k+1) =\left[\begin{array}{c}
>     x_1(k); \\
>     x_2(k); \\
>     \delta (k);\\
>     v(k);
>     \\
>     \psi (k)
>     \end{array}\right]+\tau \left[\begin{array}{c}
>     v(k) \sin (\psi(k)); \\
>     v(k) \cos (\psi(k)); \\
>     u_1(k); \\
>     u_2(k); \\
>     v(k)\text{tan}(\delta)
> \end{array}\right]
> , y(k)=\begin{bmatrix}
>     1 & 0 &0&0&0; \\
>     0 & 1 & 0&0&0
> \end{bmatrix} x(k).\nonumber
> \end{equation}
> The source system:
> \begin{align}
> \hat{x}(k+1)= \left[\begin{array}{c}
>     \hat{x}_1(k); \\
>     \hat{x}_2(k); \\
>     \hat{x}_3 (k)
>     \end{array}\right]+\tau \left[\begin{array}{c}
>      \sin (\hat{x}_3(k)) ;\\
>      \cos (\hat{x}_3(k)) ;\\
>     u(k)
> \end{array}\right]
> , \nonumber
> \hat{y}(k)=\begin{bmatrix}
>     1 & 0 &0 ;\\
>     0 & 1 & 0
> \end{bmatrix}\hat{x}(t).
> \end{align}
> Both models are borrowed from Althof et al., "CommonRoad: Vehicle Models," IEEE Intelligent Vehicles Symposium, 2017.
>
> As the reviewer correctly mentioned, it is indeed possible to approximate or learn a system's model. However, there are two significant drawbacks to this approach. While the related literature offers some methods for computing approximate models obtained through system identification techniques, acquiring an accurate model for complex systems remains highly challenging, time-consuming, and expensive.
>
>
> Q2. The robustness of the trained models with regard to hyper parameters is unclear. An addition of ablation studies on the sensitivity for the hyper parameters could be added.
>
> A2. Thank you for the insightful comment. In our experiments, we observed that by using an over-parameterized neural network (sufficiently deep and wide), hyperparameters do not significantly affect the  training performance. We experimented with various network configurations, and the training time and convergence behavior remained consistent. Since our approach does not rely on SMT solvers for verification, as our training process is correct-by-construction, we can leverage over-parameterized networks to approximate the simulation relation and its interface function effectively.
>
> Additionally, there are no fixed values for $\eta$ and $\gamma$. Instead, our algorithm computes these values at each iteration, and if they meet the admissibility criteria—specifically, the validity conditions outlined in Assumption 9—training concludes.
>
> Even when a model is successfully identified, the subsequent steps of computing a simulation relation and an interface function are themselves very complex, as highlighted in the first part of our response. These challenges underscore the novelty and practicality of our data-driven approach.

---

> ### Author Response · Authors · 2024-11-20
> **Response to Reviewer y2Rp (part 2)**
>
> Q3.I wonder whether the authors have observed failure cases during their experiments. If so, is there a way to measure whether the failure case is due to the difficulty of learning, or due to inherent problems of the given source and target system?
>
> A3.Thank you for raising this important point—we will ensure it is addressed in the revised version. Based on our experiments, failures generally occur when the source and target systems are fundamentally unrelated. For example, when we applied our method using a DC motor as the source system and an inverted pendulum as the target system, we were unable to identify a simulation relation capable of guaranteeing output closeness.
>
> Another instance of failure arose when we employed under-parameterized neural networks that lacked sufficient capacity to effectively approximate the simulation relation and its corresponding interface function.
>
> It is also important to note that our method is sound, meaning it provides a sufficient condition as long as the training converges. However, the inability to find a simulation relation and interface function using our approach does not imply that such a relation does not exist.
>
> Q4. Nit picks
>
>
> A4. We appreciate the reviewer's comment. We have resolved minor typographical errors throughout the paper.
>
> We sincerely thank the reviewer for their detailed analysis and thoughtful feedback on our method. We appreciate the time and effort you have dedicated to reviewing our paper and will incorporate your suggestions to enhance its clarity and quality.
>
> We kindly request that you re-evaluate our paper in light of the clarifications and additional insights we have provided.
>
> Sincerely,
>
> Authors

---

> > ### Comment · Reviewer_y2Rp · 2024-11-29
> >
> > I thank the authors for their detailed responses.
> >
> > Response to A1: I appreciate the authors' inclusion of the more complex system example and acknowledge their contribution regarding formal guarantees of $\epsilon$-closeness for the outputs of the source and target systems. I also concur with their observation regarding the limitations of SoS methods when applied to high-dimensional systems.
> >
> > Response to A2 and A3: I value the authors' clarifications regarding the method's robustness to hyperparameter selection and their discussion of potential failure cases. While I find their claims credible, I believe these assertions would be strengthened by including supporting empirical evidence. Such data would enhance the accessibility and adoptability of the proposed method for the broader research community across various applications.
> >
> > Based on the authors' responses and my original assessment, I maintain my rating.

---

> > > ### Author Response · Authors · 2024-11-29
> > >
> > > We appreciate the reviewer’s constructive suggestions, which have helped improve our paper.
> > >
> > > If you have any further questions or concerns, please feel free to reach out.

---

### Author Response · Authors · 2024-12-02
**Author Rebuttal**

We would like to thank the reviewers for providing detailed comments that have helped to improve the quality of our manuscript. We have provided rebuttals to the comments of each reviewer.

Summary of changes:

1. New Higher-Dimensional Case Study:

We have introduced a new case study involving higher-dimensional systems. Specifically, the source system is a 3-dimensional car model, while the target system is a 5-dimensional car model. The target system is described by the following equations:
\begin{equation}
    x(k+1) =\left[\begin{array}{c}
    x_1(k); \\
    x_2(k); \\
    \delta (k);\\
    v(k);
    \\
    \psi (k)
    \end{array}\right]+\tau \left[\begin{array}{c}
    v(k) \sin (\psi(k)); \\
    v(k) \cos (\psi(k)); \\
    u_1(k); \\
    u_2(k); \\
    v(k)\text{tan}(\delta)
\end{array}\right]
, y(k)=\begin{bmatrix}
    1 & 0 &0&0&0; \\
    0 & 1 & 0&0&0
\end{bmatrix} x(k).\nonumber
\end{equation}
The source system is defined as:
\begin{align}
\hat{x}(k+1)= \left[\begin{array}{c}
    \hat{x}_1(k); \\
    \hat{x}_2(k); \\
    \hat{x}_3 (k)
    \end{array}\right]+\tau \left[\begin{array}{c}
     \sin (\hat{x}_3(k)) ;\\
     \cos (\hat{x}_3(k)) ;\\
    u(k)
\end{array}\right]
, \nonumber
\hat{y}(k)=\begin{bmatrix}
    1 & 0 &0 ;\\
    0 & 1 & 0
\end{bmatrix}\hat{x}(t).
\end{align}
Both models are borrowed from Althof et al., "CommonRoad: Vehicle Models," IEEE Intelligent Vehicles Symposium, 2017.

2. Limitations Section:

In response to reviewer feedback, we have added a dedicated limitations section that clearly outlines the shortcomings of our method.

3. Discussion on State-of-the-Art Comparison:

We have also included a discussion section to clarify why no comparison with state-of-the-art methods is provided. To the best of our knowledge, this is the first work that aims to establish a simulation relation and its interface function between two given systems in a data-driven manner.

4. Clarity Improvements:

We have addressed the reviewers’ concerns regarding clarity by rewriting parts of the paper and correcting typographical errors.

---

> ### Comment · Reviewer_59oi · 2024-12-02
> **Response**
>
> Dear Authors,
>
> I gave you the lowest rating. However, I did not plan to change it based on the current version and contributions. I hope my rating will not discourage your future work so that I will share a few more suggestions below.
>
> 1. A high-dimensional or low-dimensional example is not a problem. Most physical systems are high-dimension and thus intractable. So, the order reduction is used to create the trackable and analyzable models for analysis and design. The essential job is to guarantee the experimental example you set has a small gap with reality.
>
> 2. This paper's topic is related to control. A significant study of control is to propose a new framework to address real, important problems and, more importantly, implement and demonstrate the control framework in real systems. You may not have the real systems, but you can consider advanced simulators with solid records for transferring to the real systems. Directly using the numerical and toy examples from some publications is not a good way; your papers may face similar antagonistic comments. Your example is a vehicle or a robotic system. So, I suggest a few simulators: Gazebo, CarSim (not free), and Isaac Sim. Building the platform for your experiment is not easy, but it will be very worthwhile.
>
> 3. Compared with the claim of 'this paper is the first one to propose ....,' I prefer the claim like `the proposed framework is to address Challenges ...' Some people may have similar ideas to you but do not publish the paper and become the first, just because they think the framework is not practical due to strong assumptions, etc.

---

> > ### Author Response · Authors · 2024-12-02
> >
> > We thank the reviewer for their suggestions; however, the submission deadline for the final version was November 28th.
> >
> >
> > We have acknowledged that the sample complexity of our approach is exponential with respect to the state dimension of the system. This is a common challenge for data-driven methods that provide formal guarantees and is not unique to our approach.
> >
> >
> > Additionally, we respectfully disagree with the reviewer’s assertion that experiments must always be realistic, even in control-related research. We have provided examples to support this point, including those from established control venues such as HSCC, CDC, LCSS, and TAC.

---

> ### Comment · Reviewer_59oi · 2024-12-02
> **Response**
>
> Authors,
>
> You can read papers from CDC, LCSS, and TAC. Most of them are purely theoretical papers; they have very, very strong theoretical contributions. In this way, the experiment requirement for them can be lowered. How do you define your paper? A theory or practical paper? From a theoretical point of view, your paper cannot be compared with theirs, with almost zero theoretical contribution. From the implementation point of view, your paper cannot be compared with a robotics paper, like the ones published in IROS. Lastly, do you think your paper has fundamental contributions to ML?

---

> > ### Comment · Reviewer_59oi · 2024-12-02
> >
> > To my knowledge, TAC is a top-tier and reputable control journal. CDC, LCSS (ACC), and HSCC are good conferences, but their acceptance bars are much lower than ICLR.

---

> > > ### Author Response · Authors · 2024-12-02
> > >
> > > Our paper bridges the gap between neural networks and formal guarantees—two areas that traditionally do not intersect easily. Numerous works on neural certificates have been published in top-tier machine learning venues, including:
> > >
> > > Zhang et al., "Exact Verification of ReLU Neural Control Barrier Functions," NeurIPS 2024.
> > >
> > > Meit et al., "ControlSynth Neural ODEs: Modeling Dynamical Systems with Guaranteed Convergence," NeurIPS 2024.
> > >
> > > Abate et al., "Neural Abstractions," NeurIPS 2022.
> > >
> > > Chang et al., "Neural Lyapunov Control," NeurIPS 2019.
> > >
> > > Yang et al., "Lyapunov-stable Neural Control for State and Output Feedback: A Novel Formulation," ICML 2024.
> > >
> > >
> > > We respectfully disagree with the reviewer’s characterization that our contribution is "almost zero." We have clearly outlined our contributions in both the paper and our responses to your comments.

---

> > > > ### Comment · Reviewer_59oi · 2024-12-02
> > > >
> > > > Are you sure the claim "gap between neural networks and formal guarantees—two areas that traditionally do not intersect easily"? I suggest using Google Scholar to search.

---

> > > > > ### Author Response · Authors · 2024-12-02
> > > > >
> > > > > We have provided examples of neural certificates published in top-tier machine learning venues with similar settings and assumptions, and we respect the reviewer’s right to their opinion.

---

### Meta-Review · Area_Chair_NsjN · 2024-12-20

**Metareview:**

This paper proposes to achieve a formal simulation relation between two discrete-time, continuous-state dynamical control systems. The goal is to provide formal guarantees on transfer learning settings.

The paper tackles an important problem and the authors have shown that they are happy to improve on the listed weaknesses. However, the main weaknesses regarding comparison with relevant baselines, further evidence towards robustness of the method, and strengthening the contribution either theoretically or practically remain. The authors are furthermore encouraged to look into approaches to safe transfer in reinforcement learning. The authors should also notice that the revised version exceeds the page limit, which means that the paper would require a thorough editing step.

**Additional Comments On Reviewer Discussion:**

The discussion was engaging, and the authors and one reviewer could not come to an agreement as to how to judge works in the control or ML community. However, this discussion was not decisive in the decision to reject. The authors are encouraged to improve the above mentioned points.

---

### Decision · Program_Chairs · 2025-01-22

Reject